# Electroreduction of unactivated alkenes using water as hydrogen source

Yanwei Wang[1], Qian Wang[1], Lei Wu[2], Kangping Jia[1], Minyan Wang [2] ✉ & Youai Qiu [1] ✉

Herein, we report an electroreduction of unactivated alkyl alkenes enabled by [Fe]-H, which is provided through the combination of anodic iron salts and the silane generated in situ via cathodic reduction, using $H_2O$ as an H-source. The catalytic amounts of Si-additive work as an H-carrier from $H_2O$ to generate a highly active silane species in situ under continuous electrochemical conditions. This approach shows a broad substrate scope and good functional group compatibility. In addition to hydrogenation, the use of $D_2O$ instead of $H_2O$ provides the desired deuterated products in good yields with excellent D-incorporation (up to >99%). Further late-stage hydrogenation of complex molecules and drug derivatives demonstrate potential application in the pharmaceutical industry. Mechanistic studies are performed and provide support for the proposed mechanistic pathway.

Electrochemistry, employing electrons as intrinsically safe and sustainable redox reagents represents an eco-friendly strategy in organic synthesis[1–18]. In this regime, electrochemical hydrogenation has gained increasing attraction due to the availability of renewable electrical energy, and cost savings obtained from replacing thermal, stoichiometric chemical reductants, and high pressure of $H_2$ inputs. In recent years, elegant work has been reported that low-cost and user-friendly $NH_3$, $H_2O$, alcohol, and DMSO have been employed as H-sources in the electrochemical hydrogenation of carbon-carbon double bonds. However, most of these works are limited to activated olefins, such as styrene derivatives and/or acrylates with more positive reduction potentials[19–31]. Contrastingly, as it is more difficult to directly reduce unactivated alkyl alkenes ($E_{1/2} < −3.0$ V vs. SCE) than activated olefins (for instance: methyl acrylate, $E_{1/2} = −2.10$ V vs. SCE; styrene, $E_{1/2} = −2.58$ V) due to their more negative reductive potentials[32], strategies for the electrochemical hydrogenation of unactivated alkyl alkenes have proven elusive[33,34]. Therefore, the development of a general strategy for electrochemical hydrogenation of unactivated alkyl alkenes is still highly desirable and remains challenging.

Transfer hydrogenation (TH) has emerged as a promising and powerful strategy due to its simple operation and is much safer than direct hydrogenation using hydrogen gas, since gas containment or pressure vessel is required[35–39]. And various reagents including silanes, boranes, acids and alcohols etc. have been used as hydrogen sources in the TH processes of alkenes[40–46]. However, there are a few examples using low-cost, non-toxic and eco-friendly water as the hydrogen donor. Formidable challenges remain in the transformations: high bond dissociation energy in $H_2O$ (118 kacl $mol^{-1}$)[47]; delicate metal-complexes are necessary (Fig. 1A). And it allows easy isotopic labeling of products by employing economical $D_2O$ in replacement of $H_2O$ as deuterium source during the TH process of alkenes, which could be a straightforward approach to acquire α,β-deuterated alkyl compounds with much cost saving. And the resulting deuterated products are crucial in the field of pharmaceutical science and mechanistic studies[48–53]. Elegant works of $H_2O$ activation, including oxidative addition activation and coordination-induced bond weakening, using transition metal complexes or main group elements with empty $d$ or $p$ orbitals have been disclosed[54]. However, direct proton transfer (PT) from water for further hydrogenation still remains challenging (Fig. 1B). Notably, thiol mediated PT/HAT using water as H-source have been investigated in the field of photocatalysis[55], which limiting its application in metal catalyzed transfer hydrogenation might due to the poison effect on metal catalysts. Hence, the development of methodologies for the highly

[1]State Key Laboratory and Institute of Elemento-Organic Chemistry, Frontiers Science Center for New Organic Matter, College of Chemistry, Nankai University, 94 Weijin Road, Tianjin 300071, China. [2]State Key Laboratory of Coordination Chemistry, School of Chemistry and Chemical Engineering, Nanjing University, Nanjing 210023, China. ✉e-mail: wangmy@nju.edu.cn; qiuyouai@nankai.edu.cn

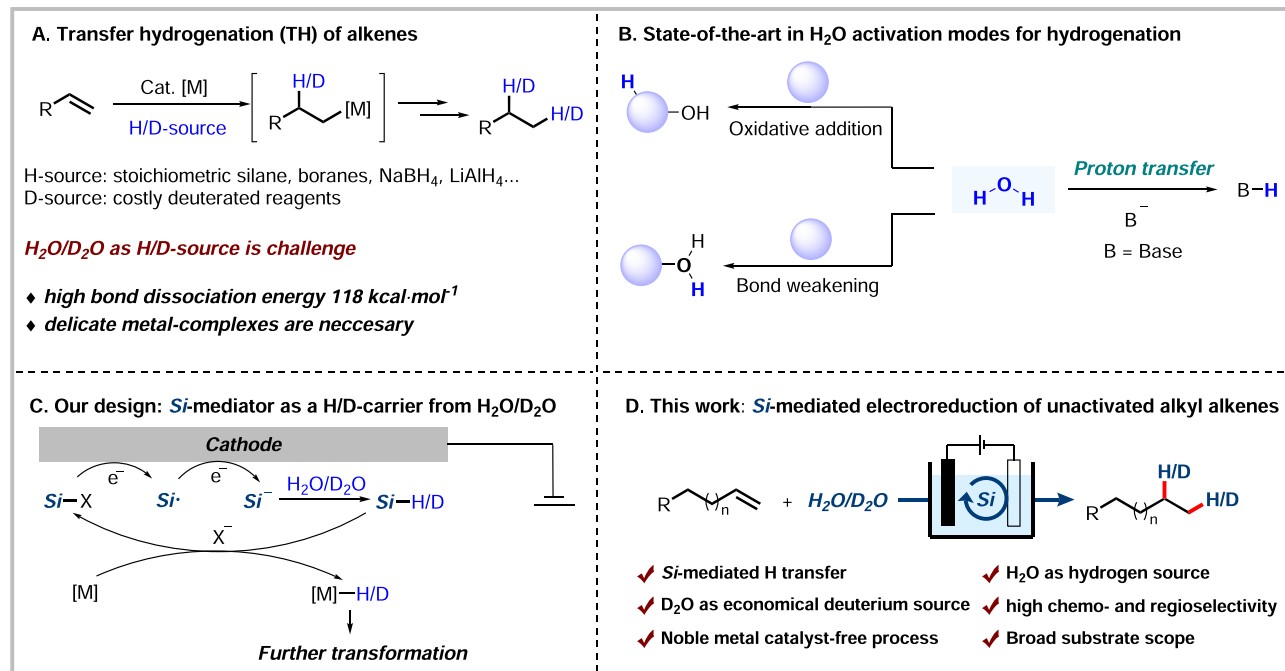

**Fig. 1 | Strategies for the hydrogenation of alkenes. A** Transfer hydrogenation (TH) of alkenes. **B** State-of-the-art in $H_2O$ activation modes for hydrogenation. **C** Our design: Si-mediator as a H/D-carrier from $H_2O/D_2O$. **D** This work: Si-mediated electroreduction of unactivated alkyl alkenes.

efficient hydrogenation employing $H_2O/D_2O$ as H/D-source of alkyl alkenes remains critical.

Meanwhile, silane is a highly active H-source and is widely used in organic synthesis, especially in combination with transition-metal catalysts in the TH reactions[56–58]. However, in most cases, excess amounts of silane are required, which limits its practical application. Chlorosilane could be activated under electroreductive conditions through two rounds of single-electron transfer to give silyl anions for further transformation[59,60]. Inspired by the electroreductive events of chlorosilane and our continuous interest in electroreductive transformations[61–65], we envisioned that chlorosilane could be used as an H/D-carrier under electroreductive conditions, which could in situ generate a key silane intermediate with $H_2O/D_2O$. Then, the Si-species could then engage in the next catalytic cycle after TH with transition metal under the electrochemical conditions (Fig. 1C).

Herein, we report an electroreduction of unactivated alkyl alkenes promoted by a catalytic amount of simple chlorosilane using $H_2O/D_2O$ as the H/D-source (Fig. 1D). Notable features of this strategy include: (a) using a simple and low-cost catalytic amount of chlorosilane as an H-carrier in the reaction system; (b) using $H_2O/D_2O$ as the economical H/D-source; (c) noble transition-metal free, low-cost Fe performing dual function as the anode and in situ generation of metal hydride; (d) high chemoselectivity and regioselectivity; (e) late-stage hydrogenation of naturally occurring compounds and drug derivatives. Detailed mechanistic insights provided strong support for the pivotal role of a H-carrier in the transformation.

## Results
### Optimization of electroreductive hydrogenation
Initially, we employed 4-(pent-4-en-1-yloxy)−1,1′-biphenyl (**1**) as the model substrate and examined various reaction conditions under a constant current for the electroreductive hydrogenation of unactivated alkyl alkenes. After careful optimization, hydrogenation product **2** was obtained in 92% NMR yield under a constant current (6 mA) in DMF with $Ph_3SiCl$ as a catalyst, $H_2O$ as an H-source, $TBABF_4$ as a supporting electrolyte, Fe plate as the anode and Ni foam as the cathode (Table 1, entry 1). The control experiment showed that $Ph_3SiCl$

was essential for this transformation as its omission led to very low efficiency (entry 2). Several anodic materials were investigated (entries 3−5), while Fe(+) was proven to be optimal. Different cathodic materials, such as Fe plate or GF, could also be used, affording the desired product in 90% and 75% yield, respectively (entry 6). Next, when TMSCl or TMSBr was used as H-carriers, the desired product was provided in good yields as well (entry 7). It was found that $(TMS)_2O$ could also promote this transformation with high efficiency (entry 8). When $Ph_3SiH$ was employed as an additive in the reaction, it gave the product in 73% yield (entry 9). In addition, the effect of the solvent on the transformation was investigated, including DMF, DMA, NMP, and $CH_3CN$. As a result, DMF was found to be optimal for this transformation (entry 10). Employing MeOH or EtOH as the H-source provided the desired product in high yield (entry 11), however, we chose low-cost and readily available $H_2O$ as the preferred H-donor in the transformation. Inspired by previous elegant reports that involved the use of 3d transition metals, such as Co or Ni, for carbon-halogen bond activation via an SET process[66–72], we envisaged that catalytic amounts of Co- or Ni- complexes could behave as a mediator to promote the cleavage of the Si–Cl bond. Indeed, when $CoBr_2.dtbpy$ or $NiBr_2.dtbpy$ was added to the reaction system, the unactivated alkene substrate could be consumed completely, and the transformation worked in very high efficiency at 5 mA giving the product with 98% isolated yield (entries 12−14). Finally, the control experiment demonstrated the essential role of electricity, as no product could be detected in its absence (entry 15). We chose entry 14 as the optimized electrochemical hydrogenation condition.

### Substrate scope
With the optimized reaction conditions established, we next explored the scope and generality of the reaction, as shown in Fig. 2. Initially, various representative terminal alkenes were proven suitable, affording the corresponding hydrogenation products in good to high yields. To our delight, the length of the alkene chain showed no significant influence on the efficiency of the transformation (from 4C to 10C) except only a slight decrease in the yield when the chain became longer (**2−8**, 90−98%). Next, terminal alkenes derived from phenols

**Table 1 | Screening of reaction conditions.[a]**

Reaction scheme: 1 → 2 under Fe(+)/Ni Foam(−), 6 mA, Ph₃SiCl (20 mol%), H₂O (3.0 equiv.), TBABF₄, DMF, RT, Ar.

| Entry | Variation from standard conditions | Yield of 2 (%) | Recovery of 1 (%) |
|---|---|---|---|
| 1 | None | 92 | 6 |
| 2 | w/o Ph$_3$SiCl | trace | 81 |
| 3 | Al(+)/Ni Foam(−) | 12 | 73 |
| 4 | Zn(+)/Ni Foam(−) | 0 | 95 |
| 5 | Mg(+)/Ni Foam(−) | 0 | 93 |
| 6 | Fe or GF as cathode | 90/75 | 5/10 |
| 7 | TMSCl or TMSBr instead of Ph$_3$SiCl | 89/86 | 0/0 |
| 8 | (TMS)$_2$O instead of Ph$_3$SiCl | 90 | trace |
| 9 | Ph$_3$SiH instead of Ph$_3$SiCl | 73 | 14 |
| 10 | DMA, NMP or CH$_3$CN as solvent | 10/20/0 | 87/30/90 |
| 11 | MeOH or EtOH instead of H$_2$O | 93/91 | 0/0 |
| 12 | with CoBr$_2$·dtbpy (5 mol%) | 94 | 0 |
| 13 | with NiBr$_2$·dtbpy (5 mol%) | 95 | 0 |
| **14[b]** | **with NiBr$_2$·dtbpy (5 mol%)** | **99(98)[c]** | **0** |
| 15 | no electricity | 0 | >99 |

Bold formatting shows that entry 14 is the optimal reaction condition.

*dtbpy* 4,4'-di-tert-butyl-2,2'-bipyridine, *TBABF₄* ⁿBu₄NBF₄, *RT* room temperature, *GF* graphite felt, *TMSCl* chlorotrimethylsilane, *TMSBr* bromotrimethylsilane, *(TMS)₂O* hexamethyldisiloxane, *DMF* N, N-dimethylformamide, *DMA* N, N-Dimethylacetamide, *NMP* N-methyl-2-pyrrolidone.
[a] Reaction conditions: undivided cell, Fe as the anode, Ni foam as the cathode, constant current = 6 mA, 1 (0.3 mmol, 1.0 equiv.), Ph₃SiCl (20 mol%), H₂O (3.0 equiv.), TBABF₄ (1.0 equiv.) in DMF (4.0 mL), room temperature, 10 h, under Ar atmosphere. Yields were determined by ¹H NMR spectroscopy using 1,3,5-trimethoxybenzene as the internal standard.
[b] Constant current = 5 mA.
[c] Isolated yield.

bearing different substituents were examined, including those with electron-rich or deficient groups, which all worked smoothly under the standard conditions and furnished the corresponding hydrogenation products in good to high yields (**10–21**). Simple alkenes without O-linkage in the alkene chain gave an almost quantitative product (**9**, 98%). Then, several N-containing substrates were tested, and to our delight, the carbazole, indole, and aniline motifs were all impregnable under the electrochemical conditions (**22–25**). Furthermore, terminal alkenes containing ester linkages derived from simple acids or alcohols proceeded well in the transformation (**26–30**). Notably, free carboxyl and hydroxyl groups remained intact under the standard conditions, demonstrating good functional group compatibility in this electro-reduction process. (**31** and **32**).

More importantly, this hydrogenation process showed excellent regioselectivity under standard conditions. When two types of carbon-carbon double bonds were present in the same molecule, the reaction occurred preferentially on the monosubstituted terminal alkene, as demonstrated with products **33** and **34** in Fig. 2. Polysubstituted alkenes showed no conversion under the standard conditions, which might be attributed to more steric hindrance, so in situ generated Ph₃SiH suppresses the hydrogenation process. Therefore, we speculated that a smaller silane could promote the hydrogenation process. Conceptually, the addition of catalytic amounts of PhSiH₃ (30 mol%) improved the efficiency of the transformation with polysubstituted alkenes to a large extent, including branch alkenes (**35** and **38**), internal alkenes (**36** and **37**) and cycloalkenes (**39** and **40**). To further demonstrate the general application of this methodology, simple commercially available straight-chain aliphatic alkenes were treated under our electroreductive conditions and gave the desired hydrogenation product in moderate to good yields (**41–44**). Moreover, this protocol could enable the hydrogenation of simple styrenes in moderate to high yields, but conjugated diene worked in very low efficiency

under the standard conditions (for details, please see the Supplementary Information).

The electrochemical hydrogenation was further expanded to the late-stage application of complex compounds (Fig. 3). Alkenes derived from drugs (including ibuprofen, flurbiprofen, menthol), α-amino acid derivatives, (including L-valine, L-methionine, and L-phenylalanine), natural compounds, (such as progesterone, estrone and cholesterol), and carbohydrates, (such as fructose, and galactose) were all amenable to the electroreductive hydrogenation transformation, providing the desired products in moderate to good yields (**45–58**). These results indicated that this strategy offered a potential synthetic method for the selective hydrogenation of complex compounds and drug molecules under mild conditions.

To our delight, this strategy could enable the electroreductive deuteration of alkyl alkenes using D₂O as the economical deuterium source under slightly modified reaction conditions, affording the deuterated products in good yields with excellent D-incorporation (up to >99%, Fig. 4). Simple terminal alkenes derived from phenols bearing different substituents worked smoothly under the electroreductive deuteration conditions and gave the desired products in moderate to good yields with excellent D-incorporation (**59–70**). N-containing substrates, such as carbazole, indole and aniline derivatives, were also tolerant under the electroreductive conditions providing the desired products in moderate to excellent D-incorporation (**71–74**). These preliminary results demonstrated that this strategy could offer a potential synthetic method for the convenient introduction of deuterium to organic compounds under mild conditions.

## Mechanistic studies

To gain insight into the mechanism of this transformation, several mechanistic experiments were conducted. When substrate **1** was treated under the standard reaction conditions, GC-MS analysis

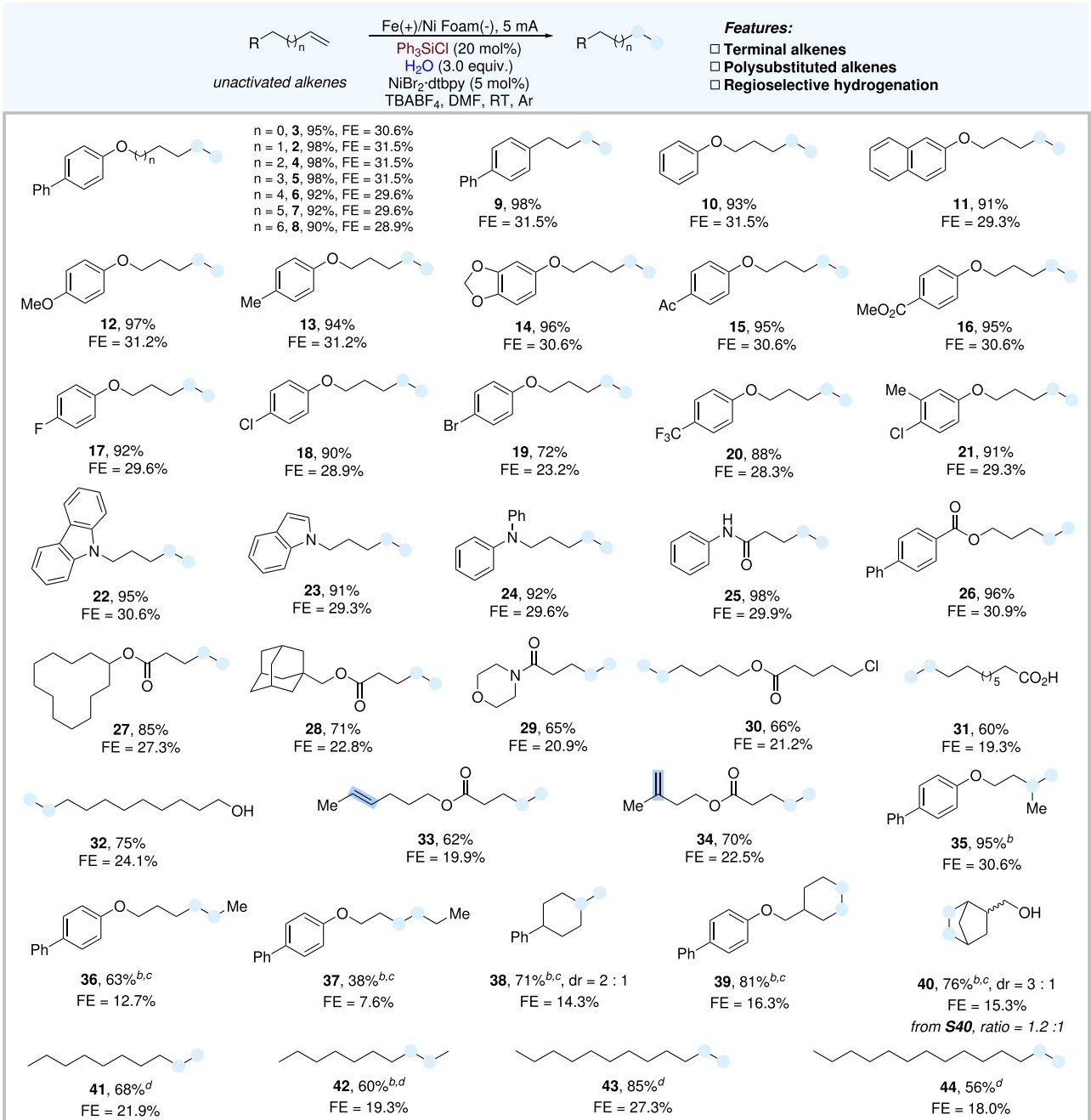

**Fig. 2 | Scope of electroreductive hydrogenation.** Reaction conditions: [a]Fe as anode, Ni Foam as cathode, constant current = 5 mA, unactivated alkene (0.3 mmol, 1.0 equiv.), NiBr$_2$·dtbpy (5 mol%), Ph$_3$SiCl (20 mol%), H$_2$O (3.0 equiv.), TBABF$_4$ (1.0 equiv.) in DMF (4.0 mL), room temperature, 10 h, under Ar atmosphere. Isolated yields. FE faradaic efficiency. [b]Ph$_3$SiCl (10 mol%), PhSiH$_3$ (30 mol%). [c]8 mA. [d]Yields were determined by GC.

detected the presence of Ph$_3$SiH (Fig. 5a). Control experiments suggested that Ph$_3$SiH could promote the electroreductive hydrogenation process in good efficiency (Fig. 5b). Thus, we speculated that Ph$_3$SiH might be an active intermediate in the catalytic cycle. Moreover, alkyl silicon compound (**75**) was synthesized and treated under the standard conditions and no hydrogenation product through C−Si bond cleavage was detected (Fig. 5c). In addition, a control experiment using a catalytic amount of **75** in replacement of Ph$_3$SiCl was conducted, and no hydrogenation product was detected (Fig. 5d). These results showed that the activation of alkenes through the formation of C−Si bond in the transformation could be ruled out.

Next, the deuteration experiments of **S3** with 5.0 equivalents of PhMe$_2$SiD with/without Ni-complex gave desired product **3**-D in 90%

and 88% yields, respectively, with D-incorporation (α/β: 85%/79% and 56%/85%) (Fig. 5e). The deuterated ratio was inverse in the two experiments, which may be attributed to the regioselectivity of different metal hydrides (Ni−H vs. Fe−H). And DFT calculations indicated that the NiH complex preferred to form the branch intermediate with alkenes, which then underwent protonation (Supplementary Fig. 9). For the FeH complex, the regioselectivity was diametrically opposed (Supplementary Fig. 10). These results of DFT calculations were in line with our experimental studies. These results illustrated that Ni-catalyzed cycle was involved in this transformation as well. Combined with the conclusion drawn from Fig. 5a, the catalytic amount of Si-catalyst could work as an H-carrier from H$_2$O/D$_2$O to alkenes. Moreover, when internal alkenes (**S36** and **S37**) were treated with 20

**Fig. 3 | Scope of late-stage modification of biorelevant compounds.** Reaction conditions: [a]Fe as anode, Ni Foam as cathode, constant current = 5 mA, unactivated alkene (0.3 mmol, 1.0 equiv.), NiBr$_2$·dtbpy (5 mol%), Ph$_3$SiCl (20 mol%), H$_2$O (3.0 equiv.), TBABF$_4$ (1.0 equiv) in DMF (4.0 mL), room temperature, 10 h, under Ar atmosphere. Isolated yields. FE faradaic efficiency. [b]50 °C.

equiv. of D$_2$O, it gave α/β/γ- and α/β/γ/δ-position deuteration products **36**-D and **37**-D in 53% and 35% yields, respectively (Fig. 5f, g). According to these results, we presumed that the Ni-complex could promote the chain-walking process to provide the terminal alkenes and then hydrogenation occurred, as no product could be detected in its absence.

Thereafter, a commonly used substrate for ring-opening experiments, ((1R,2S)−2-vinylcyclopropyl)benzene (**76**), was employed in the electroreductive hydrogenation system in the absence of the Ni-complex. The ring-opening product (**77**) and hydrogenation product (**78**) were detected by ¹H NMR spectroscopy (Fig. 5h). According to literature reports[73–78], 3d metal salts could react with silane to generate [M]-hydride species, which have a tendency to undergo migration insertion processes with alkenes (providing the intermediate **I** or **I′**). We speculated that the ring-opening product was formed through β-carbon elimination of intermediate **I**, which was followed by protonation to give product **77** (Fig. 5i). Intermediate **I′** could then transform into hydrogenation to give product **78**.

In order to gain further insight into the reaction mechanism, several cyclic voltammetry (CV) experiments were performed. CV experiments on Ph$_3$SiCl gave the reductive peaks at $E_{p/2}$ = −2.4 V vs. Ag/Ag⁺ in DMF under Ar atmosphere (Supplementary Fig. 16). As shown in

Fig. 5D, we systemically investigated the behaviors of TMSCl and TMSBr via ¹⁹F NMR tests. We found TMSCl/TMSBr could be mostly transformed into TMSF in very short time even without electricity, which was attributed to the reactions between TMSCl/TMSBr and TBABF$_4$. The characteristic ¹⁹F NMR signal of TMSF could be detected in 3 min in the reaction system (13% for TMSCl, 14% for TMSBr vs. 20%, 4-fluoroanisole as internal standard), and there was no significant decline after stirred for 12 h (14% for TMSCl, 14% for TMSBr). These results indicated highly active chlorosilane/bromosilane could exist in the reaction system in the form of fluorosilane. And fluorosilane could efficiently promote the transformation as well (Supplementary Table 2, entry 4). As shown in Fig. 5E, the mixture of nickel catalyst and ligand exhibited two reduction peaks at −1.82 V and −2.55 V vs. Ag/Ag⁺ (Fig. 5E, red line), corresponding to the reduction of Ni$^{II}$/Ni$^I$ and Ni$^I$/Ni$^0$, respectively. When CV experiments of the nickel catalyst and ligand were conducted in the presence of increasing equivalents of Ph$_3$SiCl, it was observed that the reduction current of Ni$^{II}$/Ni$^I$ is positively correlated with the amount of Ph$_3$SiCl, and the sign for the generation of Ni$^0$ complex is significantly enhanced. These results suggested that both the Ni$^{II}$-ligand complex and Ph$_3$SiCl underwent reduction reactions at the cathode and the in-situ generated Ni$^0$ complex appears to be the active catalyst[79].

**Fig. 4 | Scope of electroreductive deuteration.** Reaction conditions: $^a$ Fe as anode, Cu as cathode, constant current = 10 mA, unactivated alkenes (0.3 mmol, 1.0 equiv.), NiBr$_2$·dtbpy (5 mol%), Ph$_3$SiCl (20 mol%), D$_2$O (20.0 equiv.), TBABF$_4$ (1.0 equiv.) in DMF (4.0 mL), room temperature, 10 h, under Ar atmosphere. Isolated yield. FE faradaic efficiency. D-inc. % determined by $^1$H NMR.

Based on the mechanistic experiments and CV studies, a plausible mechanism for this electrochemical hydrogenation of unactivated alkenes was proposed (Fig. 5F). Iron salts were continuously generated at the anode, and the catalytic cycle was initiated by cathodic reduction of chlorosilane to provide silyl radical intermediate **A**. Then, intermediate **A** underwent the second one-electron reduction to give silyl anion intermediate **B**. Upon protonation of **B** with H$_2$O, a silane intermediate C is formed. The generated iron species **Fe-INT1A** on the anode react with silane through a σ-bond metathesis reaction, resulting in the formation of hydrogenated iron species **Fe-INT2A**, and regenerating the halosilane to complete the silicon catalytic cycle. Subsequently, intermediate **Fe-INT2A** undergoes a migration insertion process with alkene substrate **1**, leading to the formation of linear intermediate **Fe-INT3A** and branched intermediate **Fe-INT3B**. Both intermediates further undergo protonation with water, resulting in the production of hydrogenation products **2** and the generation of hydroxy iron species **Fe-INT4A**. Following a σ-bond metathesis reaction between **Fe-INT4A** and silicon hydride in the system, silanol **D** is formed, which remains stable in the system and can undergo gradual fluorination to re-enter the silicon catalytic cycle[80]. Notably, employing D$_2$O instead of H$_2$O could provide the electroreductive deuteration product. According to the mechanistic studies, a Ni-catalyzed cycle was involved in the transformation as well. And we have elucidated the Ni-catalyzed catalytic cycle as a minor pathway in the Supplementary Information (Supplementary Figs. 8, 9 and 11).

As showed in Table 1, Zn(+) or Mg(+) could not enable this transformation under the standard conditions, which was in line with the proposed [Fe]-H involved mechanism. However, when Zn(+) or Mg(+) was employed combined with the deposited Ni Foam cathode (coated by Fe particles after the first round electrolysis) under the standard conditions, provided the hydrogenation products in 83% and

37% yields respectively (Supplementary Table 11, entries 4 and 7). These results showed that the heterogeneous electrochemical hydrogenation process proceeded on the coated Ni Foam cathode might be involved in the transformation as well. (For more details about the heterogeneous process, please see the Supplementary Information)

## Discussion

In conclusion, we report an efficient and facile electroreductive hydrogenation/deuteration of unactivated alkyl alkenes promoted by a catalytic amount of simple chlorosilane using H$_2$O/D$_2$O as the H/D source. In this protocol, low-cost Fe performed dual function: Fe worked as anode and the iron salts generated in situ behaved as the high active [Fe]-catalyst to promote the hydrogenation process. The Si-additive worked as an H-carrier from water to in situ generate a highly active silane under electroreductive conditions, which then combined with [Fe]-catalyst to form the [Fe]–H species for further hydrogenation. This approach showed broad substrate scope and good functional group compatibility. In addition to hydrogenation, employing D$_2$O instead of H$_2$O could provide the electroreductive deuterated products with good yields and excellent D-incorporation (up to >99%). Further late-stage carboxylation of biorelevant molecules demonstrated its potential application in the pharmaceutical industry. Mechanistic studies provided strong support for the key role of chlorosilane as a H-carrier. Further applications of the electroreductive hydrogenation strategy are currently underway in our laboratory.

## Methods

### General procedure of electroreductive hydrogenation of unactivated alkenes

The electrolysis process was carried out in an undivided cell with a Fe plate anode (10 mm × 20 mm × 0.10 mm) and a Ni Foam cathode

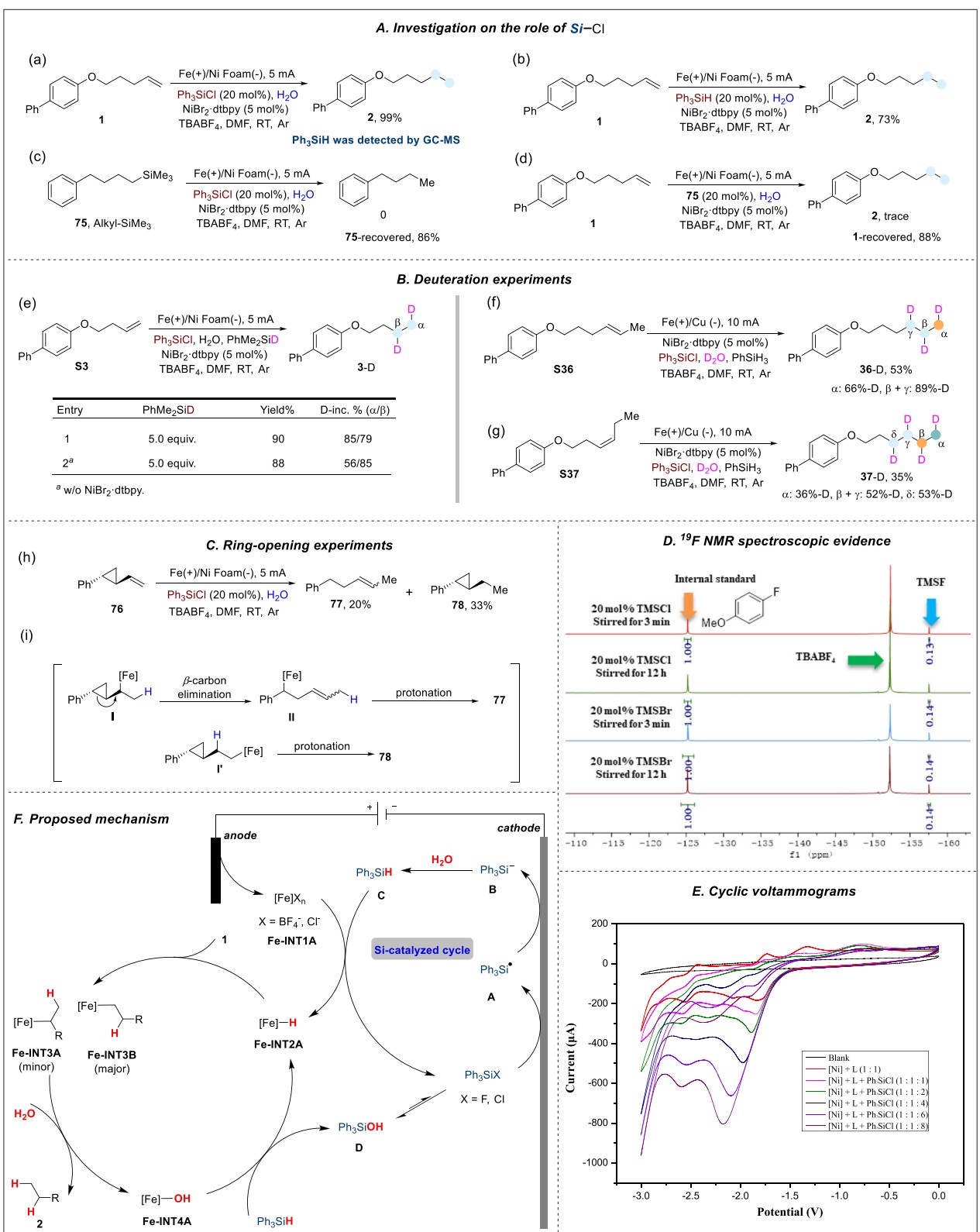

**Fig. 5 | Preliminary mechanistic studies. A** Investigation on the role of chlorosilane. **B** Deuteration experiments. **C** Ring-opening experiments. **D** $^{19}$F NMR spectroscopic evidence for the formation of fluorosilanes. **E** CV experiments, using glass carbon as the working electrode, Pt plate and Ag/Ag$^+$ as the counter and reference electrodes. Scan rate: 100 mV s$^{-1}$. DMF/$^n$Bu$_4$NBF$_4$ (0.1 M). [Ni] = NiBr$_2$·dme, L = 4,4′-di-tert-butyl-2,2′-bipyridine. **F** Proposed mechanism.

(10 mm × 20 mm × 0.30 mm). To a 15 mL oven-dried undivided electrochemical cell equipped with a magnetic bar was added unactivated alkene (0.30 mmol, 1.0 equiv.), NiBr$_2$.dtbpy (7.3 mg, 0.015 mmol, 5 mol %), Ph$_3$SiCl (17.7 mg, 0.06 mmol, 20 mol%), H$_2$O (16.2 mg, 0.9 mmol, 3.0 equiv.) and $^n$Bu$_4$NBF$_4$ (98.8 mg, 0.30 mmol, 1.0 equiv.) under Ar atmosphere. The electrolysis process was performed at 5.0 mA of constant current for 10 h at room temperature. After that, the electrodes were washed with EtOAc (3 × 5 mL) in an ultrasonic bath. H$_2$O (20 mL) was added to the organic system, and the resulting mixture was extracted with EtOAc (3 × 20 mL) and the combined organic phase was washed with brine, dried by anhydrous MgSO$_4$, filtered, and concentrated in vacuo. The crude product was purified by column chromatography to furnish the desired product.

### General procedure of electroreductive deuteration of unactivated alkenes

The electrolysis process was carried out in an undivided cell with a Fe plate anode (10 mm × 20 mm × 0.30 mm) and a Cu plate cathode (10 mm × 20 mm × 0.30 mm). To a 15 mL oven-dried undivided electrochemical cell equipped with a magnetic bar was added unactivated alkene (0.30 mmol, 1.0 equiv), NiBr$_2$.dtbpy (7.3 mg, 0.015 mmol, 5 mol %), Ph$_3$SiCl (17.7 mg, 0.06 mmol, 20 mol%), D$_2$O (120 mg, 6 mmol, 20 equiv.) and $^n$Bu$_4$NBF$_4$ (98.8 mg, 0.30 mmol, 1.0 equiv.) under Ar atmosphere. The electrolysis process was performed at 5.0 mA of constant current for 10 h at room temperature. After that, the electrodes were washed with EtOAc (3 ×5 mL) in an ultrasonic bath. H$_2$O (20 mL) was added to the organic system, and the resulting mixture was extracted with EtOAc (3 × 20 mL) and the combined organic phase was washed with brine, dried by anhydrous MgSO$_4$, filtered, and concentrated in vacuo. The crude product was purified by column chromatography to furnish the desired product.

### Gram-scale synthesis of 2

The electrolysis process was carried out in an undivided cell with a Fe plate anode (5 cm × 2.5 cm × 0.5 mm) and a Ni Foam cathode (5 cm × 2.5 cm × 0.5 mm). To a 200 mL oven-dried undivided electrochemical cell equipped with a magnetic bar was added unactivated alkene (10 mmol, 1.0 equiv), NiBr$_2$.dtbpy (242 mg, 0.5 mmol, 5 mol%), Ph$_3$SiCl (600 mg, 2 mmol, 20 mol%), H$_2$O (540 mg, 30 mmol, 3.0 equiv.) and $^n$Bu$_4$NBF$_4$ (3.29 g, 10 mmol, 1.0 equiv.) under Ar atmosphere. The electrolysis process was performed at 40 mA of constant current for 26 h at room temperature. After that, the electrodes were washed with EtOAc in an ultrasonic bath. H$_2$O (20 mL) was added to the organic system, and 20 mL of 1 N HCl aq. was carefully added after the completion of the reaction to help the removal of the excess scrap iron from the reaction system, and the resulting mixture was extracted with EtOAc and the combined organic phase was washed with brine, dried by anhydrous MgSO$_4$, filtered, and concentrated in vacuum. The crude product was purified by column chromatography to furnish the desired product (**2**) in 81% (1.94 g) yield.

## Data availability

The authors declare that the data supporting the findings of this study are available within the article and its Supplementary Information files. Extra data are available from the author upon request. Source data are provided with this paper.

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

## Acknowledgements

Financial support from National Key R&D Program of China (2022YFA1503200) (Y.Q.), National Natural Science Foundation of China (Grant No. 22371149, 22188101) (Y.Q.) and (Grant No. 22301144) (Y.W.), the Fundamental Research Funds for the Central Universities (No. 63223015) (Y.Q.), Frontiers Science Center for New Organic Matter, Nankai University (Grant No. 63181206) (Y.Q.) and Nankai University are gratefully acknowledged

## Author contributions

Y.Q. supervised the project, and provided guidance on the project. Y.Q. and Y.W. conceived and designed the study and wrote the manuscript. Y.W., Q.W. and K.J. performed the experiments, mechanistic studies and revised the manuscript, L.W. and M.W. conducted the DFT calculation and revised the manuscript. Y.W., Q.W. and L.W. contributed equally to this work. All authors contributed to the analysis and interpretation of the data.

## Competing interests

The authors declare no competing interest.
