## [Peer Review File · Nature Communications]

REVIEWER COMMENTS

Reviewer #1 (Remarks to the Author):

The authors report on the electrocatalytic hydrogenation of unactivated alkenes, using chlorosilanes as mediators and H₂O as the source of hydrogen atoms. The presented research is also placed in the relevant context of late stage functionalization of pharmaceutically relevant compounds.

Both, electrohydrogenations—particularly with water—and late stage functionalization of complex organic molecules are topical research areas of high importance. I evaluate the manuscript to be novel and interesting to a broad readership. Yet, in addition to some minor comments, I have multiple concerns with several of the claims and the presented control experiments, which I like to elaborate in the following.

In their introduction, the authors highlight that previously reported electro-hydrogenation schemes rely on metal-hydride intermediates that are not water stable, as a big drawback for using H₂O as the source of hydrogen in those systems.

In their own work, however, the authors claim to generate silane intermediates with reactive Si-H functions, which are likewise not water stable. In conjunction with the fact that the faradaic efficiency of their process is relatively low (ranging between 8 and 32%), this “selling point” is misleading and it should be clarified that despite the generally interesting aspects of novel silane mediated e-hydrogenation the problem of water incompatible intermediates is still remains.

The faradaic efficiency should be referred to in the main text, as it is an important aspect to evaluate the actual applicability of the proposed synthesis method. In the current version, faradaic efficiencies are only presented in the supporting information in table S-8. I recommend to include this information together with the yields in the respective substrate scope figures in the main text.

While I generally think the presented work is interesting and deserves to be published, I consider the mechanistic claims questionable and, despite a plethora of control experiments, not sufficiently supported.

As a major point of concern, I see the sensitivity of chlorosilanes towards water and DMF, which the authors even mention in their work. It seems highly unlikely that the added chlorosilane species would remain stable to participate in the proposed reaction cascade. Potentially, during the initiation

phase / in the very first cycle some chlorosilane species could be present. However, in view of the extremely low chloride concentration in the reaction solution it seems statistically extremely unlikely that the silane species would find free chloride to recover a chlorosilane in relevant quantity during catalysis. This should be addressed and changed.

To shed light on this aspect, the resting state / equilibrium species into which the chlorosilane is converted to should be identified for further clarification of the mechanistic cycle.

In the same context, On page 12, lines 206-261 in the manuscript, the authors state “Concurrently, chlorosilane was regenerated in the catalytic cycle under continuous electroreductive conditions”. I highly doubt that chlorosilane is ever regenerated during the cycle. Hence, I like to ask the authors to either prove or remove this statement.

With respect to their control experiment shown in the SI, table S2 entry 7, it seems that hydroxysilanes, which I consider the most likely intermediate in a potential homogeneous catalytic cycle, are active but less so than chlorosilane precursors. How do the authors rationalize this in the context that chlorosilane precursors are expected to immediately convert by reaction with H₂O or DMF?

My further concerns regard the specific role of Fe ions, liberated from the sacrificial Fe anode during the constant current electrolysis experiments, for the observed hydrogenation activity.

Assuming the rest of the authors' proposed homogeneous electro-hydrogenation mechanism is in principle correct, the following control reactions should work and I would like to see it performed to convince me of the claimed reaction cascade:

a) The reaction conditions of Figure 5 Scheme E, can be mimicked without applied electric current, simply by mixing catalytic amounts of Fe(BF₄)₂ (or Fe(BF₄)₃, depending on what the solution state Fe species oxidation state is supposed to be?), and stoichiometric amounts of alkene substrate with stoichiometric amounts (or excess) of Ph₃Si-H. Should the proposed catalytic cycle be correct, then stoichiometric amounts of alkane should be produced. If this does not work, potentially water should be added after some reaction time, because hydrolysis of metal-alkyl bonds might be an important reaction step that is not captured in the authors presented mechanism, but it seems very plausible.

Should neither of the conditions produce alkane in high yield, I would argue the proposed mechanism is not correct and potentially the authors are dealing with heterogeneous catalysis instead (see below).

b) Additionally, the role of the NiBr₂(dtbpy) as proposed co-catalyst can be checked by a control reaction in which this is also added under otherwise similar conditions as described above. Given that according to Table 1 entries 1 and 13, the addition of NiBr₂(dtbpy) is irrelevant to the yield within experimental error (92 % vs. 95%), this control experiment could shed further light on the importance of the additive.

The presence of nickel complexes could, in principle, follow a reaction cascade as mentioned in this recent publication: Asian J. Org. Chem. 2023, 12, e202200590, which the authors may also want to mention in their discussion.

Also, the publication on stoichiometric hydrogenation of alkenes with silanes and H₂O and Fe(acac)₃ should be considered in the mechanistic discussions (J. Am. Chem. Soc. 2020, 142, 19316.).

Under the assumption that the authors are electro-generating hydrosilanes as claimed, likely the mechanisms presented in the manuscripts above might be operational.

Alternatively a heterogeneous hydrogenation pathway, which is well known for Fe/Ni mixed metal anodes, could be operational (see J. Chem. Educ. 2004, 81, 1350. and references therein). By electrochemical deposition of Fe particles on the Ni cathode, such reactivity could be obtained. The authors should discuss and evaluate this option with the following control experiment:

a) After a performed electro-hydrogenation under their standard conditions, the presence of Fe particles on the nickel working electrode should be probed. Suitable methods can include GI-PXRD, Raman spectroscopy, XPS analysis or SEM / TEM – EDX mapping to detect deposited Fe.

b) A working electrode that was potentially coated with Fe particles after a first electro-hydrogenation run under standard conditions should be re-used in fresh electrolyte – now with a different anode material (like Zn or Mg), to see if the yield for alkene hydrogenation is much improved compared to new / cleaned nickel working electrodes in conjunction with Zn or Mg anodes.

As a final general aspect, I find it problematic to propose specific reactive species in the catalytic cycle when applying constant current electrolysis, given that resulting electrode potentials are constantly changing, and given that cell geometry, immersion depth of electrodes, different pairs of WE and CE material, water content, etc. may drastically change working electrode potentials from run to run and over time. This means different active species and different reaction pathways may be enabled unknowingly from run to run. The authors should discuss this issue in their manuscript.

In fact, this may be the explanation for the, in part, difficult to understand trends reported for several control experiments in the supporting information, which I would like to address in the following:

1) Table S1 seems to have the 6mA entry as a clear outlier. It seems illogical that the FE goes up for that one data point, whereas all other points together show a linear trend for higher yield with higher constant current applied.

2) Table S3: How do the authors rationalize these trends? Why should more chlorosilane decrease the yield, particularly in the context of their proposed mechanism?

3) Table S6 is very confusing to me and requires further discussion and/or confirmation. How can “no H₂O added” produce 92% yield, while 1 and 2 equiv. H₂O added produce no hydrogenated product at all, then 3 equiv. H₂O produce again exactly 92% yield, and finally 5 equiv. H₂O again produce no product at all.

I absolutely doubt this data and the authors must provide solid proof and explanation for this experimental outcome, should it not have been experimental error.

4) There is a second table “S6...”

5) I must note that I have not checked and interpreted all NMR spectra of the many reported compounds, which was also not the highest priority for me in view of the many other aspects I raised above. Regardless, I like to point out that the NMR data are presented in a way that makes it hardly possible to see all multiplets clearly, due to wide sections of blank baseline and huge distance between the peak labels and the actual peaks in the chosen presentation style. Maybe the authors can improve data clarity by presenting their NMR spectra in more appropriate ppm ranges for each compound and with peak labels closer to the peaks?

In summary, I think the general aspect of clearly demonstrated electro-hydrogenation of unactivated alkenes with H₂O and the many demonstrated substrates are interesting and relevant to be published. I think the mechanistic claims are not sufficiently supported by the experimental data, and some control experiments, such as the dependence on water content are very difficult to rationalize. Other control experiments, such as the role of nickel co-catalysts seem over interpreted. In general, constant current electrolysis makes reproducibility of results dependent on the cell design, which makes mechanistic claims even more difficult. I trust the isolated yields the authors present and therefore their observed hydrogenation activity, but I cannot support a publication of this work unless more solid proof for the mechanism is presented, or unless the mechanistic claims are made

much less specific to reflect the facts that at in its current form the manuscript does not even investigate the possibility of hereogeneous pathways, the actualy steady state species of the silane during catalysis, or specifics of the Fe ions role during catalysis.

Without more detailed mechanistic support, I consider a more specialized journal more suitable for the presented work. Should the raised points of concern be addressed thoroughly, then I consider Nature Communications a suitable journal to publish the generally interesting work.

Reviewer #2 (Remarks to the Author):

The manuscript submitted by Qiu, Wang, and co-workers reports an elegant electrocatalytic system to achieve hydrogenation and deuteration of unactivated alkenes. The method does not require the use of noble metals and uses inexpensive and catalytic amounts of chlorosilanes or hydrosilanes. The substrate scope is broad with good functional group tolerance and the system is applicable to a variety of complex molecules. Detailed mechanistic studies and control experiments are given, elucidating the electroreduction and subsequent hydrogenation process. Sufficient data and information are provided in the supporting information and allow for hands-on replication.

My only concern with this work is the complexity of the reaction system as dual metal is involved in the process together with multiple electrochemical steps. Meanwhile, the reaction is also quite sensitive to the chlorosilane/water loading (SI Table S3 and S6). These could potentially compromise their practical utilizations. Details points and questions are summarised below:

1. TMSBr seems to be a very sensitive silane source. While it is likely, as the author mentioned, to be stabilized by DMF. Is there any experimental evidence showing the stability of TMSCl or TMSBr in this system (e.g., concentration at different hold times)?
2. (TMS)₂O shows 90% yield which is very interesting. What would be the mechanism when using (TMS)₂O? Was it also reduced cathodically?
3. Figure 2, for some of the internal alkenes, an extra 30% of PhSiH₃ is added. In this case, I assume more hydrates would promote the reactivity. What if the loading of Ph₃SiCl be increased to 40%, or 40% PhSiH₃ was used in lieu of any chlorosilanes? It would be interesting to see any reactivity difference.
4. Since Ph₃SiH is the key intermediate in the system, could the system be developed by using hydrosilanes (e.g. Ph₃SiH or Ph₂SiH₂ or PhSiH₃) as a single catalyst and use electrochemistry to turn

over the catalytic cycle? This might help reduce the catalyst loading or may promote new transformations.

5. On the CV, Ni complex was shown to promote the reduction of chlorosilanes in DMF. I am wondering if iron ions could have similar effects.

6. Ni is also known to react with hydrosilanes to form Ni-H species. In this regard, could Ni be used as a single metal catalyst to achieve the same chemistry? A control with 20% Ni(II) might be a good start to verify.

7. In SI tables S3 and S6, I am surprised that the loading of Ph₃SiCl and water could have such a dramatic difference in reactivity. Especially for water, it seems 0 and 3 equiv works but all other equivalents shut down the reactivity. Is there any explanation for the observed reactivity? For instance, the 0 equiv water gives a 92% yield as H could come from DMF but 1 equiv water shows 0% percent yield which is hard to rationalize.

8. Given the sensitivity of the materials loading, it would be good for the authors to demonstrate a scale-up (e.g. 1~5 g) to further demonstrate the utility of this work.

Reviewer #3 (Remarks to the Author):

The hydrogenation of alkene has always been an important research topic for organic chemists. In this manuscript, the authors described an electrochemical hydrogenation of unactivated alkyl alkenes facilitated by catalytic amounts of simple chlorosilanes using H₂O as the H-source, which is a new reduction approach. The reaction showed a broad substrate scope and good functional group compatibility. Mechanism studies showed that the Si-mediator worked as an H-carrier from H₂O to generate a highly active silane species in situ under continuous electrochemical conditions. This work provides a green and selective hydrogenation pathway for alkenes, and has potential application in the pharmaceutical industry. It is worthy of publication in Nature Communications.

Some questions:

1. Only alkyl alkenes were employed in this work. I want to know if conjugated alkenes and aryl alkenes are suitable for this reaction.

2. In SI, in the optimization of reaction conditions section, for the effect of the H₂O loading (Table S-6), if 2 or 5 equiv. H₂O was added, the yield was 0; but in the presence of 3 equiv. H₂O, a high yield of 92% was obtained. It is very surprising. And also, a 92% yield was given in the absence of H₂O. What is the hydrogen source here? These results should be discussed and explained.

3. Can the authors make an analysis for the valence of the iron salts produced on the anode?

The point-by-point response to the reviewers' comments

Reviewer 1

Question 1: In their introduction, the authors highlight that previously reported electrohydrogenation schemes rely on metal-hydride intermediates that are not water stable, as a big drawback for using H₂O as the source of hydrogen in those systems.

In their own work, however, the authors claim to generate silane intermediates with reactive Si-H functions, which are likewise not water stable. In conjunction with the fact that the faradaic efficiency of their process is relatively low (ranging between 8 and 32%), this "selling point" is misleading and it should be clarified that despite the generally interesting aspects of novel silane mediated e-hydrogenation the problem of water incompatible intermediates is still remains.

Response: Thank you for the professional comments. We agree the comments from the reviewer, we have removed the corresponding discussion "and the key organometallic intermediates might be incompatible with excess water" in the introduction and the relevant description "organometallic intermediates are incompatible with water" in Figure 1 from the manuscript, in case of misleadingness.

Question 2: The faradaic efficiency should be referred to in the main text, as it is an important aspect to evaluate the actual applicability of the proposed synthesis method. In the current version, faradaic efficiencies are only presented in the supporting information in table S-8. I recommend to include this information together with the yields in the respective substrate scope figures in the main text.

Response: Thank you for the suggestion. The relevant faradaic efficiencies of the transformations have been added to the main text (Figure 2, 3 and 4).

Question 3: While I generally think the presented work is interesting and deserves to be published, I consider the mechanistic claims questionable and, despite a plethora of control experiments, not sufficiently supported.

Response: We highly appreciate the reviewer's positive comments on our manuscript. To verify the proposed mechanism, we conducted a series experiments to detect the key

intermediate in the transformation, including ^{19}F NMR detection experiments and GC-MS analysis, and solid proofs have been obtained to prove our catalytic cycle. Moreover, we carried out DFT calculation to support the proposed mechanism, and corresponding details have been added to the manuscript and Supporting Information.

Question 4: As a major point of concern, I see the sensitivity of chlorosilanes towards water and DMF, which the authors even mention in their work. It seems highly unlikely that the added chlorosilane species would remain stable to participate in the proposed reaction cascade. Potentially, during the initiation phase/in the very first cycle some chlorosilane species could be present.

Response: Thanks for the comments.

Thank you for the comments. To detect the stability of chlorosilanes in this system, we systemically investigated the behaviors of chlorosilanes *via* NMR tests (using more sensitive TMSCl). We found most of TMSCl could be transformed into TMSF in very short time without electricity. The characteristic ^{19}F NMR signal of TMSF could be detected in 3 minutes in the system (13% for TMSCl *vs.* 20%, 4-fluoroanisole as internal standard), and there was no significant decline after stirred for 12 h (14%) (Figure R1-1).

Experimental details for NMR tests: All the experiments were conducted using 5 mL reaction tube on the 0.3 mmol scale at room temperature in 1.0 mL DMF (substrate **1** (0.3 mmol), TMSCl (20 mol%), $\text{NiBr}_2\cdot\text{dtbpy}$ (5.0 mol%), H_2O (3.0 equiv.), TBABF_4 (1.0 equiv.)). Then CDCl_3 and 4-fluoroanisole (0.3 mmol) were added to the system, and then took ^{19}F NMR tests.

Thus, we speculated that TMSCl could be transformed into TMSF which was stable before electrolysis in the system. And the corresponding discussion on " chlorosilane could be stabilized by DMF" have been removed from the manuscript.

Figure R1-1. ^{19}F NMR evidence of TMSF

Moreover, we conducted experiments to detect the behaviors of TMSF under the electrolysis conditions (Figure R1-2). And the results indicated that the TMSF could be consumed in the first 20 minutes. And control experiments showed fluorosilane could efficiently promote the transformation as well (Table S-2, entry 4).

Figure R1-2. ^{19}F NMR evidence of TMSF under electrolysis

Furthermore, ^{19}F NMR experiments suggested that both chlorosilane and bromosilane could convert into the corresponding fluorosilane in the system, and $(\text{TMS})_2\text{O}$, Ph_3SiH and Ph_3SiOH were stable before electrolysis (Figure R1-3).

Figure R1-3. ^{19}F NMR evidence of different Si-source (20 mol% Si-source in the reaction mixture, stirred for 2 minutes)

In conclusion, chlorosilanes could convert into corresponding fluorosilane (which was stable before electrolysis in the system) very quickly, then stepped into the following catalytic cycle under the electroreductive conditions.

Question 5: However, in view of the extremely low chloride concentration in the reaction solution it seems statistically extremely unlikely that the silane species would find free chloride to recover a chlorosilane in relevant quantity during catalysis. This should be addressed and changed.

Response: Thanks for the professional comments. We agree the comments from the reviewer, chlorosilane was not the intermediate in the reaction system, and the concentration of chlorosilane was very low even in the system before electrolysis (for

details, please see the response for Question 4). And silanol might be the key intermediate in the catalytic cycle, which remains stable in the system and can undergo gradual fluorination to re-enter the silicon catalytic cycle (*Chem. Lett.* **2015**, *44*, 1506–1508). For the proofs with respect to silanols, please see the response of the following question (Question 5). The revised catalytic was shown as below (Figure R1-4), and has been added to the manuscript as well.

Figure R1-4. The revised catalytic cycle

Question 6: To shed light on this aspect, the resting state / equilibrium species into which the chlorosilane is converted to should be identified for further clarification of the mechanistic cycle.

Response: Thanks for the comments. To investigate the Si-containing key intermediate, we detected the reaction mixture through GC-MS after electrolysis for 5 h. And a significant peak Ph_3SiOH was detected at 15.6 min (Figure R1-5), which suggesting silanols might be a stable Si-containing species in the reaction system. In order to further prove silanol was the crucial intermediate in the transformation, we employed $\text{CF}_3\text{CH}_2\text{OH}$ as H-source in lieu of H_2O to explore the behaviors of Si-catalyst in the

reaction (Employing $\text{CF}_3\text{CH}_2\text{OH}$ as H-source could promote the conversion in 90% yield as well, Figure S-5, entry 4, and the assumed intermediate $\text{CF}_3\text{CH}_2\text{OSiPh}_3$ could be detected on the ^{19}F NMR spectra).

Figure R1-5. GC-MS analysis of the reaction mixture after electrolysis

We conducted the hydrogenation of substrate **1** employing Ph_3SiCl , Ph_3SiH and Ph_3SiOH as Si-catalysts, $\text{CF}_3\text{CH}_2\text{OH}$ as H-source under the standard conditions for 2 h, then these three mixtures were detected by ^{19}F NMR. The results indicated that

$\text{CF}_3\text{CH}_2\text{OSiPh}_3$ were observed in the three reaction systems (Figure R1-6~R1-8). These results showed Si-Cl, Si-H and Si-O bonds could be activated under the standard conditions for further conversion, and silanols might be the key intermediate in the catalytic cycle when using H_2O as H-source, which undergo gradual fluorination to re-enter the silicon catalytic cycle under the developed conditions. (*Chem. Lett.* **2015**, *44*, 1506–1508).

Figure R1-6. ^{19}F NMR evidence of $\text{CF}_3\text{CH}_2\text{OSiPh}_3$ catalyzed by Ph_3SiCl (20 mol%)

Figure R1-7. ^{19}F NMR evidence of $\text{CF}_3\text{CH}_2\text{OSiPh}_3$ catalyzed by Ph_3SiH (20 mol%)

Figure R1-8. ^{19}F NMR evidence of $\text{CF}_3\text{CH}_2\text{OSiPh}_3$ catalyzed by Ph_3SiOH (20 mol%)

Question 7: In the same context, On page 12, lines 206-261 in the manuscript, the authors state “Concurrently, chlorosilane was regenerated in the catalytic cycle under continuous electroreductive conditions“. I highly doubt that chlorosilane is ever regenerated during the cycle. Hence, I like to ask the authors to either prove or remove this statement.

Response: Thank you for the suggestion. We have proved silanol might be the key intermediate in the transformation. The description "Concurrently, chlorosilane was regenerated in the catalytic cycle under continuous electroreductive condition." has been replaced by " Concurrently, silanol was generated with H₂O after H-transfer process of silane, and stepped into the next catalytic cycle." in the manuscript.

Question 8: With respect to their control experiment shown in the SI, table S2 entry 7, it seems that hydroxysilanes, which I consider the most likely intermediate in a potential homogeneous catalytic cycle, are active but less so than chlorosilane precursors. How do the authors rationalize this in the context that chlorosilane precursors are expected to immediately convert by reaction with H₂O or DMF?

Response: Thanks for the comments. We have provided solid experimental evidence and proved that silanol might be the crucial intermediate in the catalytic cycle in the response of Question 4 and 5.

Moreover, we have proved that chlorosilane could convert into corresponding fluorosilane in the system very quickly, which was stable before electrolysis in the system. And the corresponding discussion on " chlorosilane could be stabilized by DMF" have been removed from the manuscript. And the corresponding discussion of the conversion from chlorosilane to fluorosilane has been added to the manuscript.

Question 9: Assuming the rest of the authors’ proposed homogeneous electrohydrogenation mechanism is in principle correct, the following control reactions should work and I would like to see it performed to convince me of the claimed reaction cascade:

a) The reaction conditions of Figure 5 Scheme E, can be mimicked without applied electric current, simply by mixing catalytic amounts of Fe(BF₄)₂ (or Fe(BF₄)₃,

depending on what the solution state Fe species oxidation state is supposed to be?), and stoichiometric amounts of alkene substrate with stoichiometric amounts (or excess) of $\text{Ph}_3\text{Si-H}$. Should the proposed catalytic cycle be correct, then stoichiometric amounts of alkane should be produced. If this does not work, potentially water should be added after some reaction time, because hydrolysis of metal-alkyl bonds might be an important reaction step that is not captured in the authors presented mechanism, but it seems very plausible.

Should neither of the conditions produce alkane in high yield, I would argue the proposed mechanism is not correct and potentially the authors are dealing with heterogeneous catalysis instead (see below).

Response: Thank you for the comments.

We conducted control experiments with catalytic amounts of $[\text{Fe}]$ (5 mol%) (commercially available $\text{Fe}(\text{BF}_4)_2 \cdot 6\text{H}_2\text{O}$ and FeCl_3 ; we have tried to search for the synthetic method of $\text{Fe}(\text{BF}_4)_3$, but no relevant literatures were obtained.), 3.0 equivalent of Ph_3SiH and 1.0 equivalent of TBABF_4 in DMF, and no hydrogenation product was detected, and the unactivated alkene substrates (**1**) remained in 95% and 98% respectively (eq. 1 and 2).

Accordingly, control experiments catalyzed by $[\text{Fe}]$ (5 mol%) with the addition of 3.0 equivalent of H_2O after 6.0 h were carried out, and no hydrogenation product was detected either, and the unactivated alkene substrates (**1**) remained in 98% in these two reactions (eq. 3 and 4).

Although the hydrogenation products were not detected in the above control experiments, the mechanism we proposed could not be ruled out according to these experimental results.

(1) On the one hand, the reaction conditions during electrolysis could not be fully mimicked by simply mixing the Fe-salts and Ph_3SiH . As we all know, the concentration of catalysts and additives have great influence on the reaction. And in the optimal reaction conditions, 20 mol% Si-catalysts were employed, which means the Si-containing active species were less than 20 mol%, and stoichiometric amounts might not be benefit to this transformation. Furthermore, the concentration of Fe-salts produced on the anode which were crucial in the hydrogen transfer process could not be accurately estimated during the electrolysis. Catalytic amounts of Fe-salts might not work in the transformation.

(2) On the other hand, we speculated the electrochemical environment was indispensable in the transformation. The high active Fe-salts produced on anode could not be replaced by the additional Fe-catalyst. And active Si-species generated in situ might be the key which initiated the reaction, which leading to no product in its absence.

In summary, the conclusion drawn from the experiments which proposed by the reviewer was not completely persuasive. And the catalytic cycle we put forward could not be ruled out according to these experimental results.

Question 10: b) Additionally, the role of the $\text{NiBr}_2(\text{dtbpy})$ as proposed co-catalyst can be checked by a control reaction in which this is also added under otherwise similar conditions as described above. Given that according to Table 1 entries 1 and 13, the

addition of NiBr₂(dtbpy) is irrelevant to the yield within experimental error (92 % vs. 95%), this control experiment could shed further light on the importance of the additive.

Response: Thank you for the suggestion.

(1) Control experiments catalyzed by NiBr₂·dtbpy (5 mol%) with 3.0 equivalent of Ph₃SiH and 1.0 equivalent of TBABF₄ (or with the addition of 3.0 equivalent of H₂O after 6 h) in DMF, and no hydrogenation product was detected, and the unactivated alkyl alkene substrates (**1**) remained in 99% in these two reactions (eq. 5 and 6). According to our response of last question (Question 8 of Reviewer 1), we think the reaction conditions during electrolysis could not be fully mimicked by simply mixing the Ni-catalyst and Ph₃SiH.

- (2) Another difference between entries 1 and 13 in Table 1 was the recovery of the starting material in the two reactions (6% vs. 0%), which suggesting the addition of Ni-catalyst could accelerate the reaction. So the yields presented in entries 1 and 13 in Table 1 were not simply experimental error, and we have stated in the manuscript "when CoBr₂·dtbpy or NiBr₂·dtbpy was added to the reaction system, the unactivated alkyl alkene could be consumed completely". And the corresponding recovery of the starting material have been added to the Table 1 in the manuscript.
- (3) We conducted the control experiment using 5 mol% of NiBr₂·dtbpy in the absence of Ph₃SiCl, and no hydrogenation product was detected, and the starting substrate (**1**) was recovered in 95%. We speculated H₂O could not behave as H-source under the Ni-catalyzed conditions in this electroreductive system to form the key Ni-H intermediate, and the addition of Ph₃SiCl which transformed into Ph₃SiH *in situ* was crucial to this hydrogenation process.

(4) To shed further light on the importance of the Ni-catalyst, a series of control experiments in the absence of NiBr₂·dtbpy, and the results were displayed in the following table.

Most of the simple unactivated alkyl alkenes could work but showed low efficiency under the conditions compared with those under standard conditions. For complex compounds, hydrogenation products **46**, **51**, **56** and **57** were obtained in very low yield (25%, 37%, 12% and 25% vs. 98%, 80%, 98% and 94%); and **45**, **48**, **50** and **55** were provided in moderate yield (62%, 65%, 53% and 56%). Treated citronellol derivative under this conditions, no desired product (**54**) was detected, and decomposition of the corresponding substrate was observed after electrolysis. According to these results, NiBr₂·dtbpy played an important role in this transformation.

^aReaction conditions: Condition [A]: under standard conditions, isolated yields. Condition [B]: undivided cell, Fe as anode, Ni Foam as cathode, constant current = 6 mA, **1** (0.3 mmol, 1.0 equiv.), Ph₃SiCl (20 mol%), H₂O (3.0 equiv.), TBABF₄ (1.0 equiv.) in DMF (4.0 mL), room temperature, 10 h, under Ar atmosphere. Yields were determined by ¹H NMR spectroscopy using 1,3,5-trimethoxybenzene as the internal standard. ^b50 °C.

In summary, although the electroreductive hydrogenation could work employing chlorosilane as single catalyst, the results showed most of them worked in low efficiency compared with those with Ni-catalyst. And the addition of NiBr₂·dtbpy could efficiently suppress the decomposition of sensitive functional group and complex compounds. Thus, NiBr₂·dtbpy was a crucial additive in this transformation. The mechanism of nickel-catalyzed electroreductive of non-activated alkenes was also investigated using DFT calculations, and a detailed description has been added to the Supporting Information.

Question 11: The presence of nickel complexes could, in principle, follow a reaction cascade as mentioned in this recent publication: Asian J. Org. Chem. 2023, 12, e202200590, which the authors may also want to mention in their discussion.

Response: Thank you for the suggestion. We speculated a Ni-catalyzed cycle was involved in this transformation as well. According to the deuteration experiments in Figure 5e, the deuterated ratio was inverse in the two experiments. These results may be attributed to the regioselectivity of different metal hydrides (Ni-H vs. Fe-H). And DFT calculations indicated that the NiH complex preferred to form the branch intermediate with alkenes, which then underwent protonation (Figure S2). For the FeH complex, the regioselectivity was diametrically opposed (Figure S3). These results of DFT calculations were in line with our experimental studies. And the detailed energy profiles have been added in the revised Supporting Information.

Question 12: Also, the publication on stoichiometric hydrogenation of alkenes with silanes and H₂O and Fe(acac)₃ should be considered in the mechanistic discussions (J.Am.Chem.Soc. 2020, 142, 19316.). Under the assumption that the authors are electro-generating hydrosilanes as claimed, likely the mechanisms presented in the manuscripts above might be operational.

Response: Thank you for the suggestion. We conducted a literature survey and found that many studies reported on the catalytic cycle mediated by Fe(II)-H and Fe(III)-H species. In our system, if Fe(III)-H species are generated, they are easily reduced to Fe(II)-H species at the cathode. Therefore, we proposed a mechanism based on the Fe(II)-H species mediated catalytic cycle, and DFT calculations confirmed the energy feasibility of this proposed mechanism, see Supporting Information for details.

Question 13: Alternatively a heterogeneous hydrogenation pathway, which is well known for Fe/Ni mixed metal anodes, could be operational (see J. Chem. Educ. 2004, 81, 1350. and references therein). By electrochemical deposition of Fe particles on the

Ni cathode, such reactivity could be obtained. The authors should discuss and evaluate this option with the following control experiment:

a) After a performed electro-hydrogenation under their standard conditions, the presence of Fe particles on the nickel working electrode should be probed. Suitable methods can include GI-PXRD, Raman spectroscopy, XPS analysis or SEM / TEM – EDX mapping to detect deposited Fe.

Response: Thank you for the suggestion.

To investigate the deposits on the cathode after electrolysis under standard conditions, we conducted PXRD, Raman spectroscopy, XPS and SEM EDS mapping analysis. We carried out the electroreductive hydrogenation reaction using substrate **1** under standard conditions, then the cathode was washed by dichloromethane, then solvent and volatile were removed under vacuum for further test. And black particles could be clearly observed on the cathode (the picture of cathode after electrolysis was shown in Figure R1-9).

Figure R1-9. The cathode after electrolysis for further test

(1) We conducted the PXRD tests of Fe particles deposited on Ni Foam (Figure R1-10), but no obvious new peak was observed (so we did not carry out GI-PXRD tests), which

indicating these electrolytic deposits were amorphous.

Figure R1-10. PXRD patterns of Fe particles deposited on Ni Foam. a) Cathode after electrolysis (Cathode-2, blue line); b) Powder scraped from the surface of cathode after electrolysis (red line); c) New Ni Foam electrode (black line).

- (5) We also carried out Raman spectroscopy tests, and no signal was detected in all waveband.
- (6) XPS technique was further used to analyze the valance state of Fe particles on the cathode (Ni Foam) after electrolysis. The high resolution of spectra was shown in Figure R1-11. The fitted peaks of 706.40 and 719.50 eV indicated the existence of Fe(0), and the peaks at 709.10 and 723.20 eV corresponded to Fe(II). The peaks at 711.10 and 728.20 eV corresponded to Fe(III).

Figure R1-11. XPS spectra of deposits on the surface of Ni Foam

(7) Te SEM images indicated the morphology of deposits on the surface of the cathode was amorphous (Figure R1-12).

Figure R1-12. SEM images of deposits on the surface of Ni Foam

(8) And SEM EDS mapping revealed C, O and Fe signal localized in the deposits on the surface of Ni Foam (Figure R1-13).

Figure R1-13. SEM EDS mapping showing the C, O and Fe signal localized in the deposits on the surface of Ni Foam

In conclusion, the deposits on the surface of cathode after electrolysis were amorphous,

and C, O and Fe signal localized in the deposits. The valence of Fe particles were including Fe(0), Fe(II) and Fe(III).

Question 14: b) A working electrode that was potentially coated with Fe particles after a first electro-hydrogenation run under standard conditions should be re-used in fresh electrolyte – now with a different anode material (like Zn or Mg), to see if the yield for alkene hydrogenation is much improved compared to new / cleaned nickel working electrodes in conjunction with Zn or Mg anodes.

Response: Thank you for the comments. We conducted several parallel experiments using substrate **1** under the standard conditions, and no obvious deposits was observed in most cases (see the pictures below: Cathode-1) after electrolysis, and some black deposits was obtained in a few cases (Cathode-2). Although the morphologies of the cathode surface were not identical after the electrolysis, the reaction efficiency were stable in all of these experiments (hydrogenation products in 96~99% yields). And we found this black deposit could reacted intensely with 1 N HCl aq., and releasing gas, which suggesting the black deposits may contain Fe(0).

Cathode-1

Cathode-2

Then, we carried out a series control experiments using Cathode-1/Cathode-2 as cathode, Zn/Mg as anode, as showing in the following table. Firstly, when using Zn(+)/Ni Foam(-) as electrodes under the standard conditions, only 17% hydrogenation

product was obtained (entry 2); and using Zn(+)/Cathode-1(-) as electrodes under standard conditions, trace amount of desired product was detected (entry 3); employing Zn(+)/Cathode-2(-) as electrodes, the hydrogenation product was provided in 83% yield (entry 4). Then, control experiments using Mg as anode and Cathode-1/Cathode-2 as cathode were conducted under the standard conditions, the hydrogenation products were obtained in 0% and 37% yields (entries 6 and 7), respectively.

Entry	Anode(+)/Cathode(-)	Yield (%) ^a
1	Fe(+)/Ni Foam(-)	99
2	Zn(+)/Ni Foam(-)	17
3	Zn(+)/Cathode-1(-)	trace
4	Zn(+)/Cathode-2(-)	83
5	Mg(+)/Ni Foam(-)	0
6	Mg(+)/Cathode-1(-)	0
7	Mg(+)/Cathode-2(-)	37

^aReaction conditions: undivided cell, anode(+)/cathode(-), constant current = 5.0 mA, **1** (0.3 mmol, 1.0 equiv.), Ph₃SiCl (20 mol%), NiBr₂·dtbpy (5.0 mol%), H₂O (3.0 equiv.), TBABF₄ (1.0 equiv.) in DMF (4.0 mL), room temperature, 10 h, under Ar atmosphere. Yields were determined by ¹H NMR spectroscopy using 1,3,5-trimethoxybenzene as the internal standard.

Moreover, we conducted the electroreductive hydrogenation under conditions in the absence of NiBr₂·dtbpy (for details, please see entry 1 in Table 1 in the manuscript), and no obvious deposits was observed (see the pictures below, Cathode-3, which was similar with Cathode-1). And we employed Zn(+)/Cathode-3 as electrodes under conditions in the absence of NiBr₂·dtbpy, no desired product was obtained.

Cathode-3

In conclusion, according to the above results, we speculated the *Si*-mediated electroreductive hydrogenation process was the main path of this reaction. However, because the cathode-2 could work in high efficiency in the control experiments as well, it is impossible to fully rule out a heterogeneous process proceeded on the coated Ni Foam cathode.

Question 15: As a final general aspect, I find it problematic to propose specific reactive species in the catalytic cycle when applying constant current electrolysis, given that resulting electrode potentials are constantly changing, and given that cell geometry, immersion depth of electrodes, different pairs of WE and CE material, water content, etc. may drastically change working electrode potentials from run to run and over time. This means different active species and different reaction pathways may be enabled unknowingly from run to run. The authors should discuss this issue in their manuscript.

Response: Thank you for the comments. We understand the concerns raised by the

reviewer. And much more factors should be considered in the field of electrochemical synthesis. However, actually, for the electroreductive hydrogenation experiments conducted in our labs, good reproducibility could be observed in most cases. In order to ensure the repeatability of the experiment, we employed the reaction tubes and electrodes with the same size, and the electrodes immersed in the solution were about 1.0 cm very time. Moreover, we could learn the stability of the reaction by observing the cell voltage (Cell voltage at the begin: 1.3~1.6 V, Cell voltage at the end: 3.0~3.5 V). We did observe some cases with abnormal cell voltages (such as: the cell voltage was very low when the two electrodes were very close, or the cell voltage was very high in some cases), which leading to very low efficiencies or no reaction. But these problems could be circumvented when we re-conducted the experiments more carefully.

As for the main reaction pathway in the reaction, we have conducted much more experiments, including ^{19}F NMR detection experiments and GC-MS analysis, and solid proofs have been obtained to prove our catalytic cycle. Moreover, we carried out DFT calculation to support the proposed mechanism, and corresponding details have been added to the manuscript and Supporting Information. We speculated the Si-promoted electroreductive hydrogenation of unactivated alkyl alkenes was the main path in the transformation according to the mechanistic studies and DFT calculations. And a Ni-catalysed catalytic cycle was also involved in the conversion as a minor path. Furthermore, heterogeneous pathway was less likely but cannot be completely ruled out. The corresponding discussion has been added to the manuscript, and please see the revised Supporting Information for more details.

Question 16: 1) Table S1 seems to have the 6 mA entry as a clear outlier. It seems illogical that the FE goes up for that one data point, whereas all other points together show a linear trend for higher yield with higher constant current applied.

Response: Thank you for the comment.

We speculated that the main reasons for this result were due to: 1) the differentia of the consumed charge through the cell and 2) the effect of cell voltage under different constant current in each experiment (Decomposition of the substrate **1** or product **2** was

observed *via* C-O bond cleavage under high cell voltage, which lead to 4-Phenylphenol.). In the previous experiments (Table S-6), the consumed charge through the cell were 6.2 F mol⁻¹ (5 mA, 10 h), 7.5 F mol⁻¹ (6 mA, 10 h), 8.7 F mol⁻¹ (7 mA, 10 h), 9.9 F mol⁻¹ (8 mA, 10 h), 8.9 F mol⁻¹ (9 mA, 8 h) and 8.7 F mol⁻¹ (10 mA, 7 h), respectively. We re-conducted the experiments with the identical consumed charge (7.5 F mol⁻¹), and no obvious linear trend was observed (see the Table below). When the hydrogenation process was carried out under 8-10 mA, the final cell voltage of were much higher (6 mA, 1.3~4.2 V; 7 mA, 1.6~4.2 V; 8 mA, 2.1~4.9 V; 9 mA, 2.7~5.3 V; 10 mA, 3.3~5.6 V), and we speculated the substrate **1** or product **2** started to decompose under the higher cell voltage. And when the reaction was conducted under 6 mA, the yield and the decomposition of product (or substrate) could be well balanced. And the updated Table S-1 has been added to the Supplementary Information (Page S-7).

Table S-1: The effect of constant current.

Entry	Current (X mA)	Yield (%) ^a
1	5	71
2	6	92
3	7	91
4	8	85
5	9	83
6	10	86

^aReaction conditions: undivided cell, Fe as anode, Ni Foam as cathode, constant current = X mA, **1** (0.3 mmol, 1.0 equiv.), Ph₃SiCl (20 mol%), H₂O (3.0 equiv.), TBABF₄ (1.0 equiv.) in DMF (4.0 mL), room temperature, 10 h, under Ar atmosphere. Yields were determined by ¹H NMR spectroscopy using 1,3,5-trimethoxybenzene as the internal standard.

Question 17: Table S3: How do the authors rationalize these trends? Why should more chlorosilane decrease the yield, particularly in the context of their proposed mechanism?

Response: Thank you for the comment.

For the effect of the loading of Ph₃SiCl in Table S-3, we have re-conducted the experiments and gained the consistent result. According to Lin's work ("An Electrochemical Strategy to Synthesize Disilanes and Oligosilanes from Chlorosilanes", *Angew. Chem. Int. Ed.* **2023**, *62*, e202303592), we speculated that this result was due to the Si-Si coupling to form disilanes under the electroreductive conditions when the concentration of chlorosilane increase, and the Si-Si bond could not be reduced under the standard conditions. Furthermore, when 1.0 or 2.0 equiv. of PhMe₂SiCl was added to the reaction system, the disilane could be isolated in moderate yield, along with hydrogenation product in 20% and 15% yield, respectively. ¹H NMR (CDCl₃, 400 MHz) δ = 7.63 – 7.54 (m, 4H), 7.39 (m, 6H), 0.37 (s, 12H). ¹³C NMR (CDCl₃, 100 MHz) δ = 140.0, 133.1, 129.4, 127.9, 1.0. HRMS (ESI): Calcd for C₁₆H₂₂Si₂⁺ [M+H]⁺ 271.1333, found 271.1339.

In our mechanism, we proposed silyl anion was a possible intermediate, and this ionic intermediate could then react with another chlorosilane *via* nucleophilic substitution to afford the Si-Si coupling product when chlorosilane was in higher concentration. Furthermore, this result provides solid evidence for the proposed catalytic cycle.

Pathway of Si-Si homo-coupling:

¹H NMR spectrum of **1,1,2,2-tetramethyl-1,2-diphenyldisilane** (CDCl₃)

¹H NMR spectrum of 1,1,2,2-tetramethyl-1,2-diphenyldisilane (CDCl₃)

HRMS of 1,1,2,2-tetramethyl-1,2-diphenyldisilane (ESI)

Moreover, employing catalytic amount of disilane (Ph₃SiSiPh₃, 20 mol%) in lieu of chlorosilane, no hydrogenation of product was detected.

Therefore, the efficiency could be suppressed with chlorosilane in high concentration.

Question 18: Table S6 is very confusing to me and requires further discussion and/or confirmation. How can “no H₂O added” produce 92% yield, while 1 and 2 equiv. H₂O added produce no hydrogenated product at all, then 3 equiv. H₂O produce again exactly 92% yield, and finally 5 equiv. H₂O again produce no product at all.

I absolutely doubt this data and the authors must provide solid proof and explanation for this experimental outcome, should it not have been experimental error.

Response: Thank you for the comments.

(1) We are so sorry that we presented a series wrong data in Table S-6 when we prepared our manuscript before, and the corrected Table S-6 has been added to the Supporting Information (Page S9).

When the reaction was conducted in the absence of H₂O, 22% yield of hydrogenation product (2) was obtained, which could be attributed to the trace amount

of water in the solvent (The anhydrous DMF used in our experiments was purchased from Energy Chemical, 99.8%, Extra Dry, with molecular sieves, Water \leq 50 ppm). With the increase of the amount of water (from 0 to 3 equiv.), the yields of hydrogenation products increased; and when the addition of water was more than 3 equiv., the reaction efficiency declined.

Table S-6: The effect of the H₂O loading.^a

Entry	H ₂ O (X equiv.)	Yield (%)
1	0	22
2	1	68
3	2	81
4	3	92
5	4	83
6	5	72

^aReaction conditions: undivided cell, Fe as anode, Ni Foam as cathode, constant current = 6 mA, **1** (0.3 mmol, 1.0 equiv.), Ph₃SiCl (20 mol%), H₂O (X equiv.), TBABF₄ (1.0 equiv.) in DMF (4.0 mL), room temperature, 10 h, under Ar atmosphere. Yields were determined by ¹H NMR spectroscopy using 1,3,5-trimethoxybenzene as the internal standard.

Question 19: 4) There is a second table “S6...”

Response: Thank you for the comment. We are so sorry for the mistake. The serial number of Table S-7 and Table S-8 have been corrected in the Supporting Information (Page S-10).

Question 20: I must note that I have not checked and interpreted all NMR spectra of the many reported compounds, which was also not the highest priority for me in view of the many other aspects I raised above. Regardless, I like to point out that the NMR data are presented in a way that makes it hardly possible to see all multiplets clearly, due to wide sections of blank baseline and huge distance between the peak labels and the actual peaks in the chosen presentation style. Maybe the authors can improve data clarity by presenting their NMR spectra in more appropriate ppm ranges for each compound and with peak labels closer to the peaks?

Response: Thank you for the suggestion. We have adjusted all of NMR spectra in more appropriate ppm ranges for each compound, and we also zoomed in the multiplets to improve data clarity.

Question 21: In summary, I think the general aspect of clearly demonstrated electrohydrogenation of unactivated alkenes with H₂O and the many demonstrated substrates are interesting and relevant to be published. I think the mechanistic claims are not sufficiently supported by the experimental data, and some control experiments, such as the dependence on water content are very difficult to rationalize. Other control experiments, such as the role of nickel co-catalysts seem over interpreted. In general, constant current electrolysis makes reproducibility of results dependent on the cell design, which makes mechanistic claims even more difficult. I trust the isolated yields the authors present and therefore their observed hydrogenation activity, but I cannot support a publication of this work unless more solid proof for the mechanism is presented, or unless the mechanistic claims are made much less specific to reflect the facts that at in its current form the manuscript does not even investigate the possibility of heterogeneous pathways, the actual steady state species of the silane during catalysis, or specifics of the Fe ions role during catalysis.

Response: Thanks for the comments. We highly appreciated the reviewer's positive and professional comments, which undoubtedly improved the quality of this manuscript.

- (1) To verify the proposed mechanism, we performed a series experiments to investigate the key intermediate in the transformation, including ¹⁹F NMR detection experiments and GC-MS analysis, and solid proofs have been obtained to prove our catalytic cycle (please see the response for Questions 4, 5 and 6). Moreover, we carried out DFT calculation to support the proposed mechanism, and corresponding details have been added to the manuscript and Supporting Information.
- (2) We are so sorry that we presented a series wrong data about the amount of water in Table S-6, which leading to the confusion. And the corrected Table S-6 has been added to the Supporting Information (Page S9).
- (3) To illustrate the importance of the Ni-catalyst, a series of control experiments in the

absence of $\text{NiBr}_2 \cdot \text{dtbpy}$. Most of the unactivated alkenes could work but showed low efficiency under the conditions compared with those under standard conditions (Please see the response for Question 10). We speculated a Ni-H catalytic cycle was involved in this transformation as well. To elucidate the Ni-catalyzed catalytic cycle, we conducted corresponding DFT calculations of the Ni-cycle, and the results have been added to the Supporting Information.

- (4) Moreover, we performed the experiments to investigate the possibility of heterogeneous catalysis using cathodes coated by deposits after the first round electrolysis. And control experiments showed the coated electrode (cathode-2) could enable the hydrogenation in a few cases in high efficiency, but, actually, no obvious or few deposits could be obtained in most cases under our standard conditions. We also carried out experiments employing the cathode with few deposits (cathode-1), which promoted the transformation in very low efficiency (for details, please see the response of Question 14). These results suggest that heterogeneous pathways may be involved in the reactions but as a minor path.
- (5) Furthermore, we speculated the iron ions in the reaction system played an important role in the process of transfer hydrogenation, and the corresponding DFT calculations with respect to iron ions have been added to the manuscript and Supporting Information. Moreover, XPS analysis showed that both Fe(II) and Fe(III) could be detected in the reaction system (Figure R1-14).

Figure R1-14. XPS analysis of the iron salts produced on the anode

In summary, we speculated the Si-promoted electroreductive hydrogenation of unactivated alkenes was the main path in the transformation according to the mechanistic studies and DFT calculations. And a Ni-catalysed catalytic cycle was also involved in the conversion as a minor path. Furthermore, heterogeneous pathway was less likely but cannot be completely ruled out.

We highly appreciate the reviewer's thorough reading and comments/suggestions about our manuscript!

Reviewer 2

Question 1: TMSBr seems to be a very sensitive silane source. While it is likely, as the author mentioned, to be stabilized by DMF. Is there any experimental evidence showing the stability of TMSCl or TMSBr in this system (e.g., concentration at different hold times)?

Response: Thanks for the comments. To test the stability of TMSCl and TMSBr in this system, we systemically investigated the behaviors of TMSCl and TMSBr *via* NMR tests. We found most of TMSCl/TMSBr could be transformed into TMSF in very short

time without electricity. The characteristic ^{19}F NMR signal of TMSF could be detected in 3 min in the system (13% for TMSCl, 14% for TMSBr vs. 20%, 4-fluoroanisole as internal standard), and there was no significant decline after stirred for 12 h (14% for TMSCl, 14% for TMSBr) (Figure R2-1).

Experimental details for NMR tests: All the experiments were conducted using 5 mL reaction tube on the 0.3 mmol scale at room temperature in 1.0 mL DMF (substrate **1** (0.3 mmol), TMSCl/TMSBr (20 mol%), $\text{NiBr}_2\text{dtbpy}$ (5 mol%), H_2O (3.0 equiv.), TBABF_4 (1.0 equiv.)). Then CDCl_3 and 4-fluoroanisole (0.3 mmol) were added to the system, and took ^{19}F NMR test.

Thus, we speculated that TMSCl/TMSBr could be transformed into TMSF which was stable before electrolysis in the system. And the corresponding discussion on "chlorosilane could be stabilized by DMF" have been removed from the manuscript.

Figure R2-1. ^{19}F NMR evidence of TMSF (w/o electricity)

Question 2: $(\text{TMS})_2\text{O}$ shows 90% yield which is very interesting. What would be the mechanism when using $(\text{TMS})_2\text{O}$? Was it also reduced cathodically?

Response: Thank you for the comments.

Firstly, we conducted CV experiments of $(\text{TMS})_2\text{O}$, and the half-peak potential was observed at $E_{1/2} = -3.17 \text{ V}$ vs. Ag/Ag^+ (Figure R2-2), which demonstrated $(\text{TMS})_2\text{O}$ could be reduced under electroreductive conditions.

Figure R2-2. CV of $(\text{TMS})_2\text{O}$

Then, we conducted ^{19}F NMR tests to detect the behaviors after the cleavage of Si-O bond. We carried out the hydrogenation reaction using $(\text{TMS})_2\text{O}$ (20 mol%), $\text{NiBr}_2\cdot\text{dtbpy}$ (5 mol%), $\text{CF}_3\text{CH}_2\text{OH}$ (3.0 equiv.) as H-source and TBABF_4 (1.0 equiv.) (Employing $\text{CF}_3\text{CH}_2\text{OH}$ as H-source could promote the conversion in 90% yield as well, Figure S-5, entry 4, and the assumed intermediate $\text{CF}_3\text{CH}_2\text{OTMS}$ could be detected on the ^{19}F NMR spectra). After electrolysis for 2 h, the reaction mixture was detected by ^{19}F NMR, the characteristic signals of $\text{TMSOCH}_2\text{CF}_3$ were observed (Figure R2-3). These results indicated that $(\text{TMS})_2\text{O}$ could be transformed into $\text{TMSOCH}_2\text{CF}_3$ in the reaction system and stepped into the following catalytic cycle. Furthermore, we have proved $\text{Ph}_3\text{SiCl}/\text{Ph}_3\text{SiH}/\text{Ph}_3\text{SiOH}$ could convert to $\text{Ph}_3\text{SiOCH}_2\text{CF}_3$ (for details, please see response for Question 6 of Reviewer 1), and $\text{Ph}_3\text{SiOCH}_2\text{CF}_3$ was the key intermediate in the catalytic (Ph_3SiOH in lieu of

$\text{Ph}_3\text{SiOCH}_2\text{CF}_3$ behaved as intermediate when using H_2O as H-source). Moreover, $\text{Ph}_3\text{SiOCH}_2\text{CF}_3$ could promote this transformation in high efficiency as well (Table S-2, entry 9). Thus, the catalytic cycle was consistent with that using chlorosilane after Si-O bond cleavage.

Figure R2-3. ^{19}F NMR evidence of TMSF and $\text{TMSOCH}_2\text{CF}_3$

Question 3: Figure 2, for some of the internal alkenes, an extra 30% of PhSiH_3 is added. In this case, I assume more hydrates would promote the reactivity. What if the loading of Ph_3SiCl be increased to 40%, or 40% PhSiH_3 was used in lieu of any chlorosilanes? It would be interesting to see any reactivity difference.

Response: Thank you for the comments.

We carried out control experiments using Ph_3SiCl (40 mol%) (condition B) or PhSiH_3 (40 mol%) (condition C) for the hydrogenation of internal alkenes (S36, S37, S39 and S40). Hydrogenation products 36, 37 and 38 were not detected under conditions B and C, product 40 was obtained in 31% and trace amount respectively. These results indicating both Ph_3SiCl and PhSiH_3 were crucial in the hydrogenations of internal alkenes.

Fe(+)/Ni Foam(-), 5 mA Ph_3SiCl (20 mol%) PhSiH_3 (30 mol%) H_2O (3.0 equiv.) $\text{NiBr}_2\cdot\text{dtbpy}$ (5 mol%) TBABF_4, DMF, RT, Ar Condition A	Fe(+)/Ni Foam(-), 5 mA Ph_3SiCl (40 mol%) H_2O (3.0 equiv.) $\text{NiBr}_2\cdot\text{dtbpy}$ (5 mol%) TBABF_4, DMF, RT, Ar Condition B	Fe(+)/Ni Foam(-), 5 mA PhSiH_3 (40 mol%) H_2O (3.0 equiv.) $\text{NiBr}_2\cdot\text{dtbpy}$ (5 mol%) TBABF_4, DMF, RT, Ar Condition C	
	Condition A	Condition B	Condition C
	Condition A	Condition B	Condition C
 36	63%	0	0
 37	38%	0	0
 39	81%	0	0
 40	76%	31%	trace

Question 4: Since Ph_3SiH is the key intermediate in the system, could the system be developed by using hydrosilanes (e.g. Ph_3SiH or Ph_2SiH_2 or PhSiH_3) as a single catalyst and use electrochemistry to turn over the catalytic cycle? This might help reduce the catalyst loading or may promote new transformations.

Response: Thank you for the comments. We conducted control experiments using hydrosilane (Ph_3SiH , Ph_2SiH_2 and PhSiH_3) as single catalysts, which gave the hydrogenation product in 73%, 82% and 23% yields.

Then, Ph_2SiH_2 was employed as catalyst to conducted the hydrogenation reaction of unactivated alkenes. Some examples were displayed in the following table. Most of the unactivated alkenes showed low efficiency under the conditions compared with those under standard conditions. Treated disubstituted terminal alkene (**35**) with catalytic amount of Ph_2SiH_2 , no hydrogenation product was detected. For complex compounds, hydrogenation products **45**, **46**, **48** and **52** were obtained in very low yield (18%, 12%, 24% and 36% vs. 91%, 98%, 97% and 98%); products **50**, **54** and **58** were not detected under this condition; and **49**, **53** and **57** were provided in moderate yield (64%, 57% and 50% vs. 93%, 94% and 94%). According to these results, hydrosilane could promote the hydrogenation but in low efficiency under current conditions.

^aReaction conditions: Condition [A]: under standard conditions, isolated yields. Condition [B]: undivided cell, Fe as anode, Ni Foam as cathode, constant current = 6 mA, **1** (0.3 mmol, 1.0 equiv.), Ph₂SiH₂ (20 mol%), H₂O (3.0 equiv.), TBABF₄ (1.0 equiv.) in DMF (4.0 mL), room temperature, 10 h, under Ar atmosphere. Yields were determined by ¹H NMR spectroscopy using 1,3,5-trimethoxybenzene as the internal standard. ^b50 °C.

Question 5: On the CV, Ni complex was shown to promote the reduction of chlorosilanes in DMF. I am wondering if iron ions could have similar effects.

Response: Thank you for the comment. We conducted CV experiments of FeCl₃, and Fe^{III} displayed a reversible reduction peak at -0.52 V vs. Ag/Ag⁺ (Figure R2-4, red line). When CVs of FeCl₃ were conducted in the presence of increasing equivalents of Ph₃SiCl (blue and pink lines), no significant change of the reduction peak of Fe^{III} was

observed, which indicating iron ions have no interaction with chlorosilane.

Figure R2-4. CVs of iron ions and chlorosilanes.

Question 6: Ni is also known to react with hydrosilanes to form Ni-H species. In this regard, could Ni be used as a single metal catalyst to achieve the same chemistry? A control with 20% Ni(II) might be a good start to verify.

Response: Thank you for the professional suggestion. We conducted the control experiment using 20 mol% of NiBr₂dtbpy in the absence of Ph₃SiCl, and no hydrogenation product was detected, and the starting substrate (**1**) was recovered in 93%. We speculated H₂O could not behave as H-source under the Ni-catalyzed conditions in this electroreductive system to form the key Ni-H intermediate, and the addition of Ph₃SiCl which transformed into Ph₃SiH *in situ* was crucial to this hydrogenation process.

Question 7: In SI tables S3 and S6, I am surprised that the loading of Ph₃SiCl and water could have such a dramatic difference in reactivity. Especially for water, it seems 0 and 3 equiv works but all other equivalents shut down the reactivity. Is there any

explanation for the observed reactivity? For instance, the 0 equiv water gives a 92% yield as H could come from DMF but 1 equiv water shows 0% percent yield which is hard to rationalize.

Response: Thank you for the comments.

(1) We are so sorry that we presented a series wrong data in Table S-6 before, and the corrected Table S-6 has been added to the Supporting Information (Page S9).

When the reaction was conducted in the absence of H₂O, 22% yield of hydrogenation product (**2**) was obtained, which could be attributed to the trace amount of water in the solvent (The anhydrous DMF used in our experiments was purchased from Energy Chemical, 99.8%, Extra Dry, with molecular sieves, Water≤50 ppm). With the increase of the amount of water (from 0 to 3 equiv.), the yields of hydrogenation products increased; and when the addition of water was more than 3 equiv., the reaction efficiency declined.

Table S-6: The effect of the H₂O loading.^a

Entry	H ₂ O (X equiv.)	Yield (%)
1	0	22
2	1	68
3	2	81
4	3	92
5	4	83
6	5	72

^aReaction conditions: undivided cell, Fe as anode, Ni Foam as cathode, constant current = 6 mA, **1** (0.3 mmol, 1.0 equiv.), Ph₃SiCl (20 mol%), H₂O (X equiv.), TBABF₄ (1.0 equiv.) in DMF (4.0 mL), room temperature, 10 h, under Ar atmosphere. Yields were determined by ¹H NMR spectroscopy using 1,3,5-trimethoxybenzene as the internal standard.

(2) For the effect of the loading of Ph₃SiCl in Table S-3, we have re-conducted the experiments and gained the consistent results. According to Lin's work ("An Electrochemical Strategy to Synthesize Disilanes and Oligosilanes from Chlorosilanes", *Angew. Chem. Int. Ed.* **2023**, *62*, e202303592), we speculated that this result was due

to the Si-Si coupling to form disilanes under the electroreductive conditions when the concentration of chlorosilane increase. And when 1.0 or 2.0 equiv. of PhMe₂SiCl was added to the reaction system, the disilane could be isolated in moderate yield, along with hydrogenation product in 20% and 15% yield, respectively.

¹H NMR spectrum of 1,1,2,2-tetramethyl-1,2-diphenyldisilane (CDCl₃)

¹³C NMR spectrum of 1,1,2,2-tetramethyl-1,2-diphenyldisilane (CDCl₃)

Moreover, using catalytic amount of disilane (20 mol%) in lieu of chlorosilane, no hydrogenation of product was detected, which suggesting the Si-Si bond could not be reduced under the conditions.

Therefore, the efficiency was suppressed with chlorosilane in high concentration.

Question 8: Given the sensitivity of the materials loading, it would be good for the authors to demonstrate a scale-up (e.g. 1~5 g) to further demonstrate the utility of this work.

Response: Thank you for the comments.

We carried out this electroreductive hydrogenation of unactivated alkene (1) on 10 mmol scale under 40 mA for 26 h and gave the corresponding product in 81% yield (1.94 g), and the results have been added to the manuscript and Supporting Information.

Experimental details: The electrolysis process was carried out in an undivided cell with a Fe plate anode (5 cm × 2.5 cm × 0.5 mm) and a Ni Foam cathode (5 cm × 2.5 cm × 0.5 mm). To a 200 mL oven-dried undivided electrochemical cell equipped with a magnetic bar was added unactivated alkene (10 mmol, 1.0 equiv), NiBr₂·dtbpy (242 mg, 0.5 mmol, 5 mol%), Ph₃SiCl (600 mg, 2 mmol, 20 mol%), H₂O (540 mg, 30 mmol, 3.0 equiv.) and ^tBu₄NBF₄ (3.29 g, 10 mmol, 1.0 equiv.) under Ar atmosphere. The electrolysis process was performed at 40 mA of constant current for 26 h at room temperature. After that, the electrodes were washed with EtOAc in an ultrasonic bath. H₂O (20 mL) was added to the organic system, and 20 mL of 1 N HCl aq. was carefully added to help the removal of the excess scrap iron from the reaction system, and the resulting mixture was extracted with EtOAc (3 × 50 mL) and the combined organic phase was washed with brine, dried by anhydrous MgSO₄, filtered, and concentrated in vacuum. The crude product was purified by column chromatography to furnish the desired product (2) in 81% yield (1.94 g).

Pictures of the reaction setups for the 10 mmol scale:

We highly appreciate the reviewer's thorough reading and comments/suggestions about our manuscript!

Reviewer 3

Question 1: Only alkyl alkenes were employed in this work. I want to know if conjugated alkenes and aryl alkenes are suitable for this reaction.

Response: Thank you for the comments.

We tested four styrene derivatives, including 4-vinyl-1,1'-biphenyl, 1-vinylnaphthalene, *N*-cyclohexyl-4-vinylbenzamide and 5-vinylbenzo[*b*]thiophene, which gave the corresponding hydrogenation products in moderate to high yields (55%-95%). These results suggesting this protocol could be applied to reductive hydrogenation of styrene derivatives. And the corresponding data have been added to the Supporting Information.

4-Ethyl-1,1'-biphenyl (77)

The title compound was prepared following the general procedure, purification by column chromatography on silica gel (petroleum ether) yielded (51.8 mg, 95%) as a white solid. ¹H NMR (CDCl₃, 400 MHz) δ = 7.58 (d, *J* = 7.2 Hz, 2H), 7.52 (d, *J* = 8.2 Hz, 2H), 7.42 (t, *J* = 7.4 Hz, 2H), 7.35 – 7.24 (m, 3H), 2.70 (q, *J* = 7.6 Hz, 2H), 1.28 (t, *J* = 7.6 Hz, 3H). ¹³C NMR (CDCl₃, 100 MHz) δ = 143.5, 141.3, 138.7, 128.8, 128.4, 127.2, 127.1, 28.7, 15.7.

1-Ethylnaphthalene (78)

The title compound was prepared following the general procedure, purification by column chromatography on silica gel (petroleum ether) yielded (37.9 mg, 81%) as a colorless oil. ¹H NMR (CDCl₃, 400 MHz) δ = 8.11 (d, *J* = 8.2 Hz, 1H), 7.91 (d, *J* = 7.8 Hz, 1H), 7.76 (d, *J* = 8.0 Hz, 1H), 7.61 – 7.50 (m, 2H), 7.49 – 7.43 (m, 1H), 7.39 (d, *J* = 6.9 Hz, 1H), 3.17 (q, *J* = 7.5 Hz, 2H), 1.44 (t, *J* = 7.6 Hz, 3H). ¹³C NMR (CDCl₃, 100 MHz) δ = 140.4, 133.9, 131.9, 128.9, 126.5, 125.8, 125.5, 125.0, 123.9, 26.0, 15.2.

N-Cyclohexyl-4-ethylbenzamide (79)

The title compound was prepared following the general procedure, purification by column chromatography on silica gel (petroleum ether/EtOAc = 3:1) yielded (62.4 mg, 90%) as a white solid. ¹H NMR (CDCl₃, 400 MHz) δ = 7.67 (d, *J* = 8.1 Hz, 2H), 7.22 (d, *J* = 7.9 Hz, 2H), 6.06 (s, 1H), 4.03 – 3.89 (m, 1H), 2.67 (q, *J* = 7.6 Hz, 2H), 2.01 (d, *J* = 9.5 Hz, 2H), 1.73 (m, 2H), 1.63 (d, *J* = 10.3 Hz, 1H), 1.40 (q, *J* = 12.6 Hz, 2H), 1.23 (m, 6H). ¹³C NMR (CDCl₃, 100 MHz) δ = 166.7, 147.9, 132.6, 128.0, 127.0, 48.7, 33.4, 28.9, 25.7, 25.1, 15.5.

5-Ethylbenzo[*b*]thiophene (80)

The title compound was prepared following the general procedure, purification by column chromatography on silica gel (petroleum ether) yielded (26.7 mg, 55%) as a white solid. $^1\text{H NMR}$ (CDCl_3 , 400 MHz) δ = 7.80 (d, J = 8.3 Hz, 1H), 7.66 (s, 1H), 7.42 (d, J = 5.4 Hz, 1H), 7.29 (d, J = 5.4 Hz, 1H), 7.22 (d, J = 8.2 Hz, 1H), 2.79 (q, J = 7.6 Hz, 2H), 1.32 (t, J = 7.6 Hz, 3H). $^{13}\text{C NMR}$ (CDCl_3 , 100 MHz) δ = 140.6, 140.1, 137.3, 126.5, 125.1, 123.8, 122.4, 122.3, 29.0, 16.2.

$^1\text{H NMR}$ spectrum of 4-ethyl-1,1'-biphenyl (CDCl_3)

$^{13}\text{C NMR}$ spectrum of 4-ethyl-1,1'-biphenyl (CDCl_3)

¹H NMR spectrum of 1-ethynaphthalene (CDCl₃)

¹³C NMR spectrum of 1-ethynaphthalene (CDCl₃)

¹H NMR spectrum of *N*-cyclohexyl-4-ethylbenzamide (CDCl₃)

¹³C NMR spectrum of *N*-cyclohexyl-4-ethylbenzamide (CDCl₃)

¹H NMR spectrum of 5-ethylbenzo[*b*]thiophene (CDCl₃)

¹³C NMR spectrum of 5-ethylbenzo[*b*]thiophene (CDCl₃)

And conjugated diene, 4-(buta-1,3-dien-1-yl)-1,1'-biphenyl, was treated under the standard hydrogenation conditions, only 10% hydrogenation product was detected on the ^1H NMR. In addition, 1,4-hydrogenation product was obtained in 12% NMR yield. And the corresponding result has been added to the Supporting Information.

Question 2: In SI, in the optimization of reaction conditions section, for the effect of the H_2O loading (Table S-6), if 2 or 5 equiv. H_2O was added, the yield was 0; but in the presence of 3 equiv. H_2O , a high yield of 92% was obtained. It is very surprising. And also, a 92% yield was given in the absence of H_2O . What is the hydrogen source here? These results should be discussed and explained.

Response: Thank you for the comment.

We are so sorry that we presented a series wrong data in Table S-6 before, and the corrected Table S-6 has been added to the Supporting Information (Page S9).

When the reaction was conducted in the absence of H_2O , 22% yield of hydrogenation product (**2**) was obtained, which could be attributed to the trace amount

of water in the solvent (The anhydrous DMF used in our experiments was purchased from Energy Chemical, 99.8%, Extra Dry, with molecular sieves, Water \leq 50 ppm). With the increase of the amount of water (from 0 to 3 equiv.), the yields of hydrogenation products increased; and when the addition of water was more than 3 equiv., the reaction efficiency declined.

Table S-6: The effect of the H₂O loading.^a

Entry	H ₂ O (X equiv.)	Yield (%)
1	0	22
2	1	68
3	2	81
4	3	92
5	4	83
6	5	72

^aReaction conditions: undivided cell, Fe as anode, Ni Foam as cathode, constant current = 6 mA, **1** (0.3 mmol, 1.0 equiv.), Ph₃SiCl (20 mol%), H₂O (X equiv.), TBABF₄ (1.0 equiv.) in DMF (4.0 mL), room temperature, 10 h, under Ar atmosphere. Yields were determined by ¹H NMR spectroscopy using 1,3,5-trimethoxybenzene as the internal standard.

Question 3: Can the authors make an analysis for the valence of the iron salts produced on the anode?

Response: Thank you for the comment. To investigate the valence of the iron salts produced on the anode, we conducted XPS (X-ray photoelectron spectroscopy) analysis. We carried out the electroreductive hydrogenation reaction using substrate **1** under standard conditions, then solvent and volatile were removed from the reaction system under vacuum and detected by XPS. The high resolution spectra were shown in Figure 3-1, the binding energy of the two peaks located at 724.30 and 710.71 eV was attributed to Fe 2p_{1/2} and Fe 2p_{3/2}, respectively. The fitted peaks of 709.99 and 723.83 eV indicated the existence of Fe(II), and the peaks at 711.60 and 725.21 eV corresponded to Fe(III).

Figure 3-1. XPS analysis of the iron salts produced on the anode

We highly appreciate the reviewer's thorough reading and comments/suggestions about our manuscript!

REVIEWER COMMENTS

Reviewer #1 (Remarks to the Author):

First of all, I like to thank the authors for a very thorough work in addressing the comments of all reviewers as communicated in their response letter!

Furthermore, I like to recommend a publication of the presented work in Nature Communications, after addressing the final comments below. The reason for my further comments is an—in part—considerable discrepancy between the communication in the response letter and the performed changes to the manuscript and supporting information.

1) The authors have proven that their proposed homogeneous electro-hydrogenation pathway is active, they have further supported it with additional convincing experiments, and they have made necessary changes to the presented pathway in Figure 5F. Most importantly, the authors have proven that the actual hydrogenation is performed by an active Fe catalyst species that is in situ generated by electro-oxidation of their Fe anode, and not by a direct reaction of the alkenes with a silane / a silicon based active species. However, in multiple sections of their work, the authors make their way of communication appear as if the actual hydrogenation step was performed at an Si center and not at the Fe center. Such misleading sections include:

a) the title of the manuscript “Si-Mediated Electroreduction of Unactivated Alkenes...”

b) the abstract “we reported an electroreduction of unactivated alkyl alkenes facilitated by catalytic amounts of simple chlorosilanes using H₂O as an H-source”

c) the conclusion paragraph, where the actually active iron centers are not at all mentioned.

I consider this very misleading to readers that do not go through the entire manuscript carefully, so I like to ask the authors to make adjustments and clearly mention the role of in situ produced Fe species, in agreement with their proposed mechanism in Figure 5F, at least in the abstract and the conclusion section.

2) The reviewers did a lot of work in answering the questions 13 and 14 regarding deposition of Fe material on the Ni cathode and its re-use to check for heterogeneous hydrogenation activity. Since readers of the manuscript might also wonder about the possibility of heterogeneous electro-hydrogenation with Fe particles, I would like to see this already performed work included in the Supporting information and also referred to in the main text.

The re-using test showed that heterogeneous hydrogenation remains plausible and this should be included into the discussion and mentioned in the conclusion. Particularly the fact that control cathodes Cathode-1 and Cathode-2 that show visually very different Fe deposits afford very different hydrogenation activity when used with innocent Zn and Mg anodes, shows that such deposited heterogeneous species may have drastic effect during catalysis, depending on the specific type of deposit.

That said, I agree with the authors that their homogeneous hydrogenation pathway exists and is sufficiently active. Nevertheless, I see benefit to not “hush up” the plausible parallel hydrogenation pathway obtained by their heterogeneous Fe deposits. The findings presented in the rebuttal letter were accurately interpreted by the authors, such that both the homogeneous hydrogenation pathway and the heterogeneous hydrogenation pathway by deposited Fe species were confirmed to exist in their system. I like to ask the authors to actively communicate this in their manuscript as well, and not only in the rebuttal letter.

In fact I consider that a parallel heterogeneous hydrogenation is not at all diminishing their achievements of efficiently hydrogenating a plethora of substrates in good yields – now even with a scale-up example. However, I would consider the active choice to conceal these new findings by not reporting them a bad decision for the scientific community, given that there is much to be learned from the great efforts the authors spent in the review process.

Further minor comments:

I think in the supporting information in Table S7, the column “H-source” should be changed to “solvent”?

Reviewer #2 (Remarks to the Author):

In this revised manuscript, the authors have addressed most of the questions raised by the reviewers adequately. However, two of their responses are missing key information and would need to be clarified and modified appropriately.

First, the authors used ^1F NMR to elucidate the fate of TMSCl and TMSBr and proposed they could be quickly converted to TMSF in the reaction stream. Nevertheless, only qualitative information was given in their response and Figure 5E. It is also likely that part of the TMSCl and TMSBr was transformed to TMSF while the rest remained in the solution. While this halogen exchange process is thermodynamically favourable, TMSF is also more inert and resulted in lower reactivity compared to TMSCl or TMSBr (Table S2, entry 4 vs entry 2 or 3). I would recommend the authors to quantify the amount of TMSF in solution at different times if possible, using quant- ^1F NMR. Alternatively, TMSCl and TMSBr are detectable on GC-MS with certain instrument settings, which will also provide a better clue of the fate of TMSCl and TMSBr . ^{29}Si -NMR would also be a good alternative.

Second, in their response for Q5 for reviewer 2, CV was provided with FeCl_3 and different amount of Ph_3SiCl . It is true that there is no difference of the first reduction peak at $E_{1/2}$ at -0.52 V. However, this peak is more likely the reduction of Fe^{3+} to Fe^{2+} instead of the reduction behaviour of Ph_3SiCl . The CV experiment should be conducted by comparing CV of Ph_3SiCl with $\text{Ph}_3\text{SiCl} + \text{FeCl}_3$. Then add different equivalents of FeCl_3 to see if the reduction peak of Ph_3SiCl could be enhanced or shifted. This could better help elucidate the function of Fe^{3+} .

Apart from these, all my concerns have been addressed appropriately. I appreciate the authors' hard work on revising the manuscript and would support the publication of this work after the above-mentioned issues being addressed.

Reviewer #3 (Remarks to the Author):

The authors supplemented and revised the manuscript according to the comments of reviewers. This reviewer has no new suggestions for revision. I recommend it for publication in Nature Commun.

The point-by-point response to the reviewers' comments

Reviewer 1

Question 1: The authors have proven that their proposed homogeneous electro-hydrogenation pathway is active, they have further supported it with additional convincing experiments, and they have made necessary changes to the presented pathway in Figure 5F. Most importantly, the authors have proven that the actual hydrogenation is performed by an active Fe catalyst species that is in situ generated by electro-oxidation of their Fe anode, and not by a direct reaction of the alkenes with a silane / a silicon based active species. However, in multiple sections of their work, the authors make their way of communication appear as if the actual hydrogenation step was performed at an Si center and not at the Fe center. Such misleading sections include:

- a) the title of the manuscript "Si-Mediated Electroreduction of Unactivated Alkenes..."
- b) the abstract "we reported an electroreduction of unactivated alkyl alkenes facilitated by catalytic amounts of simple chlorosilanes using H₂O as an H-source"
- c) the conclusion paragraph, where the actually active iron centers are not at all mentioned.

I consider this very misleading to readers that do not go through the entire manuscript carefully, so I like to ask the authors to make adjustments and clearly mention the role of in situ produced Fe species, in agreement with their proposed mechanism in Figure 5F, at least in the abstract and the conclusion section.

Response: Thank you for the professional comments.

- a) The title of the manuscript "Si-Mediated Electroreduction of Unactivated Alkenes..." has been replaced by "Electroreduction of Unactivated Alkenes using H₂O as Hydrogen Source".
- b) The abstract "we reported an electroreduction of unactivated alkyl alkenes facilitated by catalytic amounts of simple chlorosilanes using H₂O as an H-source" has been changed to "we reported an electroreduction of unactivated alkyl alkenes enabled by [Fe]-H, which was provided through the combination of anodic iron salts

and the silane generated in situ *via* cathodic reduction, using H₂O as an H-source. The catalytic amounts of Si-additive worked as an H-carrier from H₂O to generate a highly active silane species in situ under continuous electrochemical conditions”.

c) The description of the active iron salts has been added to the conclusion paragraph “...In this protocol, low-cost Fe performed dual function: Fe worked as anode and the iron salts generated in situ behaved as the high active [Fe]-catalyst to promote the hydrogenation process. The Si-additive worked as an H-carrier from water to in situ generate a highly active silane under electroreductive conditions, which then combined with [Fe]-catalyst to form the [Fe]-H species for further hydrogenation....”.

Question 2: The reviewers did a lot of work in answering the questions 13 and 14 regarding deposition of Fe material on the Ni cathode and its re-use to check for heterogeneous hydrogenation activity. Since readers of the manuscript might also wonder about the possibility of heterogeneous electro-hydrogenation with Fe particles, I would like to see this already performed work included in the Supporting information and also referred to in the main text.

The re-using test showed that heterogeneous hydrogenation remains plausible and this should be included into the discussion and mentioned in the conclusion. Particularly the fact that control cathodes Cathode-1 and Cathode-2 that show visually very different Fe deposits afford very different hydrogenation activity when used with innocent Zn and Mg anodes, shows that such deposited heterogeneous species may have drastic effect during catalysis, depending on the specific type of deposit.

That said, I agree with the authors that their homogeneous hydrogenation pathway exists and is sufficiently active. Nevertheless, I see benefit to not “hush up” the plausible parallel hydrogenation pathway obtained by their heterogeneous Fe deposits. The findings presented in the rebuttal letter were accurately interpreted by the authors, such that both the homogeneous hydrogenation pathway and the heterogeneous hydrogenation pathway by deposited Fe species were confirmed to exist in their system. I like to ask the authors to actively communicate this in their manuscript as well, and not only in the rebuttal letter.

In fact I consider that a parallel heterogeneous hydrogenation is not at all diminishing their achievements of efficiently hydrogenating a plethora of substrates in good yields – now even with a scale-up example. However, I would consider the active choice to conceal these new findings by not reporting them a bad decision for the scientific community, given that there is much to be learned from the great efforts the authors spent in the review process.

Response: Thanks for comments.

The corresponding discussion on the heterogeneous hydrogenation process has been added to the manuscript “As showed in Table 1, Zn(+) or Mg(+) could not enable this transformation under the standard conditions, which was in line with the proposed [Fe]-H involved mechanism. However, when Zn(+) or Mg(+) was employed combined with the deposited Ni Foam cathode (coated by Fe particles after the first round electrolysis) under the standard conditions, provided the hydrogenation products in 83% and 37% yields respectively (Table S-11). These results showed that the heterogeneous electrochemical hydrogenation process proceeded on the coated Ni Foam cathode might be involved in the transformation as well.” And the relevant details have been added to the Supplementary Information (S107-113).

Question 3: Further minor comments:

I think in the supporting information in Table S7, the column “H-source” should be changed to “solvent”?

Response: Thanks for the comment. We are so sorry for this mistake, and the column “H-source” has been replaced by “Solvent” in Table S7.

We highly appreciate the reviewer’s thorough reading and comments/suggestions about our manuscript.

We are sure that the quality of this work will be greatly improved according to these nice comments and wise suggestions.

Reviewer 2

Question 1: First, the authors used F NMR to elucidate the fate of TMSCl and TMSBr and proposed they could be quickly converted to TMSF in the reaction stream. Nevertheless, only qualitative information was given in their response and Figure 5E. It is also likely that part of the TMSCl and TMSBr was transformed to TMSF while the rest remained in the solution. While this halogen exchange process is thermodynamically favourable, TMSF is also more inert and resulted in lower reactivity compared to TMSCl or TMSBr (Table S2, entry 4 vs entry 2 or 3). I would recommend the authors to quantify the amount of TMSF in solution at different times if possible, using quant-F NMR. Alternatively, TMSCl and TMSBr are detectable on GC-MS with certain instrument settings, which will also provide a better clue of the fate of TMSCl and TMSBr. Si-NMR would also be a good alternative.

Response: Thanks for the professional comments.

(1). We carried out the electrochemical hydrogenation reaction using 20 mol% of TMSCl, and the amount of TMSF in solution at different time have been quantified using quant-¹⁹F NMR (Figure R2-1). And the results indicated that the TMSF could be consumed in the first 20 minutes, and no TMSF was detected any more. We speculated that TMSF was very active under the electrolysis conditions, which could be quickly stepped into the following reaction in the system. Moreover, we found that fluorosilane could promote this transformation in good efficiency with low concentration.

Thus, fluorosilane might be a high active species in the reaction system.

Figure R2-1. ^{19}F NMR evidence of TMSF in the reaction system

(2). Firstly, we conducted GC-MS analysis of TMSCl and TMSBr in *n*-hexane, and the significant peaks were detected at 1.78 and 2.64 min respectively (Figure R2-2 and R2-3). Then, we investigated the reaction system (with 20 mol% TMSCl or TMSBr) using GC-MS before electrolysis, and no TMSCl (or TMSBr) was detected, and the significant peaks of TMSF were observed in both reaction system (Figure R2-4 and R2-5). These results were in line with the ^{19}F NMR experiments.

流路号:1 保留时间:1.775(扫描数:156)
 质量峰:441
 原始模式:单个 1.775(156) 基峰:92.95(8383890)
 背景模式:无 组 1 - 事件 1 Scan

Figure R2-2. GC-MS analysis of TMSCl in *n*hexane

色谱图 (All TIC) wyw-Br

色谱图 (缩放)

质谱

流路号: 1 保留时间: 2.640 (扫描数: 329)
 质量峰: 432
 原始模式: 单个 2.640 (329) 基峰: 68.95 (8391279)
 背景模式: 无 组 1 - 事件 1 Scan

Figure R2-3 . GC-MS analysis of TMSBr in "hexane

流路号:1 保留时间:1.490(扫描数:99)
 质量峰:445
 原始模式:单个 1.490(99) 基峰:77.00(748874)
 背景模式:无 组 1 - 事件 1 Scan

Figure R2-4. GC-MS analysis of TMSCl (20 mol%) in the reaction system

Figure R2-5. GC-MS analysis of TMSBr (20 mol%) in the reaction system

(3). To illustrate the fate of TMSCl and TMSBr, we conducted a series of ^{29}Si NMR experiments (Considering the boiling points of TMSCl, TMSBr or TMSF are very low, we employed Ph_3SiCl to conduct the ^{29}Si NMR experiments. And we used Et_4Si as internal standard in all of the ^{29}Si NMR spectrum.). Firstly, ^{29}Si NMR spectra of Ph_3SiCl and Ph_3SiF were reported in Figure R2-6 and R2-7, respectively. Then, we conducted the ^{29}Si NMR experiment of the reaction mixture before electrolysis (stir the reaction mixture for 10 min, then remove the solvent under vacuum for NMR test),

the results showed that Ph_3SiF was detected in the system, and no Ph_3SiCl was detected. These results were in accordance with the GC-MS analysis. Meanwhile, another two new signals were observed on the ^{29}Si NMR spectrum (Figure R2-8). According to the literature reports, DMF could react with chlorosilane to form the silyloxy derivatives (*J. Organomet. Chem.* **238**, C41–C45. (1982); *J. Org. Chem.* **49**, 1311–1312 (1984); *Eur. J. Inorg. Chem.* 5076–5080 (2006)). And two similar signals were detected in control experiments, which indicated these signals were arise from the reaction between Ph_3SiCl and DMF (Figure R2-9; dissolve Ph_3SiCl in DMF, and stir for 10 min, then remove the solvent under vacuum for NMR test.)

In conclusion, most of the chlorosilane was converted to fluorosilane in the system, and a small amount of them reacted with DMF in the system.

Figure R2-6. ^{29}Si NMR spectrum of Ph_3SiCl in CDCl_3

Figure R2-7. ^{29}Si NMR spectrum of Ph_3SiF in CDCl_3

Figure R2-8. ^{29}Si NMR spectrum of the reaction system

Figure R2-9. ^{29}Si NMR spectrum of Ph_3SiCl treated with DMF

Question 2: Second, in their response for Q5 for reviewer 2, CV was provided with FeCl_3 and different amount of Ph_3SiCl . It is true that there is no difference of the first reduction peak at $E_{1/2}$ at -0.52 V. However, this peak is more likely the reduction of Fe^{3+} to Fe^{2+} instead of the reduction behaviour of Ph_3SiCl . The CV experiment should be conducted by comparing CV of Ph_3SiCl with $\text{Ph}_3\text{SiCl} + \text{FeCl}_3$. Then add different equivalents of FeCl_3 to see if the reduction peak of Ph_3SiCl could be enhanced or shifted. This could better help elucidate the function of Fe^{3+} .

Response: Thank you for the comments. We conducted CV experiments of Ph_3SiCl , which displayed a irreversible reduction peak at -3.03 V vs. Ag/Ag^+ (Figure R2-10, red line). When CVs of Ph_3SiCl were conducted in the presence of increasing equivalents of FeCl_3 , no significant change of the reduction peak of Ph_3SiCl was observed, which indicating iron ions have no interaction with chlorosilane. The minor increases of the reduction peak of Ph_3SiCl were attributed to the superimposed effect of them.

Figure R2-10. CVs of chlorosilane with increasing amounts of FeCl_3

We highly appreciate the reviewer's thorough reading and comments/suggestions about our manuscript.

We are sure that the quality of this work will be greatly improved according to these nice comments and wise suggestions.

Reviewer 3

The authors supplemented and revised the manuscript according to the comments of reviewers. This reviewer has no new suggestions for revision. I recommend it for publication in Nature Commun.

We highly appreciate the reviewer's positive comments on our manuscript. We are sure that the quality of this work has been greatly improved according to these nice comments and wise suggestions. Thanks very much.

REVIEWERS' COMMENTS

Reviewer #2 (Remarks to the Author):

All my concerns and question have been fully addressed in the revised manuscript. I appreciate the authors for their hard work and would like to recommend publishing the manuscript in Nature Communication.

The point-by-point response to the reviewers' comments

Reviewer 2

All my concerns and question have been fully addressed in the revised manuscript. I appreciate the authors for their hard work and would like to recommend publishing the manuscript in Nature Communication.

We highly appreciate the reviewer's positive comments on our manuscript. We are sure that the quality of this work has been greatly improved according to these nice comments and wise suggestions. Thanks very much.